**Spring temperature variability over Turkey since 1800 CE reconstructed**

**from a broad network of tree-ring data**

**Nesibe Köse[(1),*], H. Tuncay Güner[(1)], Grant L. Harley[(2)], Joel Guiot[(3)]**

[(1)]Istanbul University, Faculty of Forestry, Forest Botany Department 34473 Bahçeköy-Istanbul, Turkey
[(2)]University of Southern Mississippi, Department of Geography and Geology, 118 College Drive Box 5051, Hattiesburg, Mississippi, 39406, USA
[(3)] Aix-Marseille Université, CNRS, IRD, CEREGE UM34, ECCOREV, 13545 Aix-en-Provence, France

*Corresponding author.   Fax: +90 212 226 11 13
 E-mail address: nesibe@istanbul.edu.tr

**Abstract**

The meteorological observational period in Turkey, which starts *ca.* 1930 CE, is too short for understanding long-term climatic variability. Tree rings have been used intensively as proxy records to understand summer precipitation history of the region, primarily because of having a dominant precipitation signal. Yet,, the historical context of temperature variability is unclear. Here we used higher order principle components of a network of 23 tree-ring chronologies to provide a high-resolution spring (March–April) temperature reconstruction over Turkey during the period 1800–2002. The reconstruction model accounted for 67% (Adj. $R^2 = 0.64$, $p < 0.0001$) of the instrumental temperature variance over the full calibration period (1930–2002). The reconstruction is punctuated by a temperature increase during the 20[th] century; yet extreme cold and warm events during the 19[th] century seem to eclipse conditions during the 20[th] century.. We found significant correlations between our March–April spring temperature reconstruction and existing gridded spring temperature reconstructions for Europe over Turkey and southeastern Europe. Moreover, the precipitation signal obtained from the tree-ring network (first principle component) showed highly significant correlations with gridded summer drought index reconstruction over Turkey and Mediterranean countries. Our results showed that, beside the dominant precipitation signal, a temperature signal can be extracted from tree-ring series and they can be useful proxies to reconstruct past temperature variability.

KEYWORDS: Dendroclimatology, Climate reconstruction, *Pinus nigra,* Principle component analysis, Spring temperature.

**1 Introduction**

Long term meteorological observations in the Mediterranean region allow access to 100 years of instrumental reordings of temperature, precipitation and pressure in most of the region. Moreover, natural archives as well as documentary information provide resources with which to make sensitive climate reconstructions. An extensive body of literature details climate changes in the Mediterranean region over the last two millennia (Luterbacher et al. 2012). Paleolimnological studies provide evidence that the Medieval Climatic Anomaly (MCA; 900–1300 CE) characterized warm and dry conditions over the Iberian Peninsula, while the Little Ice Age (LIA; 1300–1850 CE) brought opposite climate conditions, forced by interactions between the East Atlantic and North Atlantic Oscillation (Sanchez-Lopez et al. 2016). In addition, Roberts et al. (2012) highlighted an intriguing spatial dipole NAO pattern between the western and eastern Mediterranean region, which brought anti-phased warm (cool) and wet (dry) conditions during the MCA and LIA. The hydro-climate patterns revealed by previous investigations appear to have been forced not only by NAO, but other climate modes with non-stationary teleconnections across the region (Roberts et al. 2012).

The climate of Turkey is mainly characterized by Mediterranean macro climate (Türkeş, 1996a). Contrary to the most countries in the Mediterranean region, Turkey has relatively short meteorological records, which start in the 1930s, for understanding long-term climatic variability. On the other hand, proxy records such as speleothems (Fleitmann et al. 2009, Jex et al. 2010, Göktürk et al. 2011), lake sediments (Wick et al. 2003, Jones et al. 2006, Roberts et al. 2008, 2012, Kuzucuoğlu et al.2011, Woodbridge and Roberts 2011, Ülgen et al. 2012, Dean et

al. 2013) and tree-rings, have been used to reconstruct long term hydroclimate conditions over
Turkey.  Tree rings in particular have shown to provide useful information about the past climate
of Turkey and were used intensively during the last decade to reconstruct precipitation in the
Aegean (Griggs et al. 2007), Black Sea (Akkemik et al. 2005, 2008; Martin-Benitto et al. 2016),
Mediterranean regions (Touchan et al. 2005a), as well as the Sivas (D'Arrigo & Cullen 2001),
southwestern (Touchan et al. 2003, Touchan et al. 2007; Köse et al. 2013 ), south-central
(Akkemik & Aras 2005) and western Anatolian (Köse et al. 2011) regions of Turkey. These
studies used tree rings to reconstruct precipitation because available moisture is often found to be
the most important limiting factor that influences radial growth of many tree species in Turkey.
These studies revealed past spring-summer precipitation, and described past dry and wet events
and their duration. Recently, Cook et al. (2015) presented Old World Drought Atlas (OWDA),
which is a set of year by year maps of reconstructed Palmer Drought Severity Index from tree-
ring chronologies over the Europe and Mediterranean Basin.
Besides detailed information on precipitation history represented by these paleoscientific studies,
we have still very limited knowledge of past temperature variability of Turkey. For example,
significant decreases in spring diurnal temperature ranges (DTR) occurred throughout Turkey
from 1929 to 1999 (Turkes & Sumer 2004). This decrease in spring DTRs was characterized by
day-time temperatures that remained relatively constant while a significant increase in night-time
temperatures were recorded over western Turkey and were concentrated around urbanized and
rapidly-urbanizing cities. The historical context of this gradual warming trend in spring
temperatures is unclear. Heinrich et al. (2013) provided a winter-to-spring temperature proxy for
Turkey from carbon isotopes within the growth rings of *Juniperus excelsa* M. Bieb. since AD

1125. Low-frequency temperature trends corresponding to the end of Medieval Climatic Anomaly and Little Ice Age were identified in the record, but the proxy failed to identify the recent warming trend during the 20[th] century. In this study, we present a tree-ring based spring temperature reconstruction from Turkey and compare our results to previous reconstructions of temperature and precipitation to provide a more comprehensive understanding of climate conditions during the 19[th] and 20[th] centuries.

## 2  Data and Methods

### 2.1 Climate of the Study Area

The study area, which spans 36–42º N and 26–38º E, was based on the distribution of available tree-ring chronologies. This vast area covers much of western Anatolia and includes the western Black Sea, Marmara, and western Mediterranean regions. Much of this area is characterized by a Mediterranean climate that is primarily controlled by polar and tropical air masses (Türkeş 1996a, Deniz et al. 2011). In winter, polar fronts from the Balkan Peninsula bring cold air that is centered in the Mediterranean. Conversely, the dry, warm conditions in summer are dominated by weak frontal systems and maritime effects. Moreover, the Azores high-pressure system in summer and anticyclonic activity from the Siberian high-pressure system often cause below normal precipitation and dry sub-humid conditions over the region (Türkeş 1999, Deniz et al. 2011). In this Mediterranean climate, annual mean temperature and precipitation range from 3.6 °C to 20.1 °C and from 295 to 2220 mm, respectively, both of which are strongly controlled by elevation (Deniz et al. 2011).

2.2 Development of tree-ring chronologies

To investigate past temperature conditions, we used a network of 23 tree-ring site chronologies
(Fig. 1). Fifteen chronologies were produced by previous investigations (Mutlu et al. 2011,
Akkemik et al. 2008, Köse et al. unpublished data, Köse et al. 2011, Köse et al. 2005) that
focused on reconstructing precipitation in the study area. In addition, we sampled eight new
study sites and developed tree-ring time series for these areas (Table 1). Increment cores were
taken from living *Pinus nigra* Arnold and *Pinus sylvestris* L. trees and cross-sections were taken
from *Abies nordmanniana* (Steven) Spach and *Picea orientalis* (L.) Link trunks.

Samples were processed using standard dendrochronological techniques (Stokes & Smiley 1968,
Orvis & Grissino-Mayer 2002, Speer 2010).  Tree-ring widths were measured, then visually
crossdated using the list method (Yamaguchi 1991). We used the computer program COFECHA,
which uses segmented time-series correlation techniques, to statistically confirm our visual
crossdating (Holmes 1983, Grissino-Mayer 2001).  Crossdated tree-ring time series were then
standardized by fitting a 67% cubic smoothing spline with a 50% cutoff frequency to remove
non-climatic trends related to the age, size, and the effects of stand dynamics using the ARSTAN
program (Cook 1985, Cook et al. 1990a). These detrended series were then pre-whitened with
low-order autoregressive models to produce time series with a strong common signal and
without biological persistence. These series may be more suitable to understand the effect of
climate on tree-growth, even if any persistence due to climate might be removed by pre-
whitening. For each chronology, the individual series were averaged to a single chronology by
computing the biweight robust means to reduce the influences of outliers (Cook et al. 1990b). In
this research we used residual chronologies obtained from ARSTAN to reconstruct temperature.

The mean sensitivity, which is a metric representing the year-to-year variation in ring width
(Fritts 1976), was calculated for each chronology and compared. The minimum sample depth for
each chronology was determined according to expressed population signal (EPS), which we used
as a guide for assessing the likely loss of reconstruction accuracy. Although arbitrary, we
required the commonly considered threshold of EPS > 0.85 (Wigley et al. 1984; Briffa & Jones

140  1990).


2.3 Identifying relationship between tree-ring width and climate

We extracted high resolution monthly temperature and precipitation records from the climate
dataset CRU TS 3.23 gridded at 0.5º intervals (Jones and Harris 2008) from KNMI Climate
Explorer (http://climexp.knmi.nl) for 36–42 ºN, 26–38 ºE. The period AD 1930–2002 was
chosen for the analysis because it maximized the number of station records within the study area.

First, the climate-growth relationships were investigated with response function analysis (RFA)
(Fritts 1976) for biological year from previous October to current October using the
DENDROCLIM2002 program (Biondi & Waikul 2004). This analysis is done to determine the
months during which the tree-growth is the most responsive to temperature. RFA results showed
that precipitation from May to August and temperature in March and April have dominant
control on tree-ring formation in the area. Second, we produced correlation maps showing
correlation coefficients between tree-ring chronologies and the climate factors most important
for tree growth, which are May–August precipitation and March–April temperature, to find the
spatial structure of radial growth-climate relationship (St. George 2014, St. George and Ault
2014, Hellmann et al. 2016). For each site we used the closest gridded temperature and
precipitation values.

2.4  Temperature reconstruction

The climate reconstruction is performed by regression based on the principal component (PCs)
of the 23 chronologies within the study area. Principle Component Analysis (PCA) was done
over the entire period in common to the tree-ring chronologies.  The significant PCs were
selected by stepwise regression. We combined forward selection with backward elimination
setting $p < 0.05$ as entrance tolerance and $p < 0.10$ as exit tolerance. The final model obtained
when the regression reaches a local minimum of the root mean squared error (RMSE). The order
of entry of the PCs into the model was $PC_3$, $PC_{21}$, $PC_4$, $PC_{15}$, $PC_5$, $PC_{17}$, $PC_7$, $PC_9$, $PC_{10}$. The
regression equation is calibrated on the common period (1930–2002) between robust temperature
time-series and the selected tree-ring series. Third, the final reconstruction is based on bootstrap
regression (Till and Guiot 1990), a method designed to calculate appropriate confidence intervals
for reconstructed values and explained variance even in cases of short time-series. It consists in
randomly resampling the calibration datasets to produce 1000 calibration equations based on a
number of slightly different datasets.

The quality of the reconstruction is assessed by a number of standard statistics. The overall
quality of fit of reconstruction is evaluated based on the determination coefficient ($R^2$), which
expresses the percentage of variance explained by the model and RMSE, which expresses the
calibration error. This does not insure the quality of the extrapolation which needs additional
statistics based on independent observations, i.e. observations not used by the calibration
(verification data). They are provided by the observations not resampled by the bootstrap
process. The prediction RMSE (called RMSEP), the reduction of error (RE) and the coefficient
of efficiency (CE) are calculated on the verification data and enable to test the predictive quality
of the calibrated equations (Cook et al. 1994). Traditionally, a positive RE or CE values means a
statistically significant reconstruction model, but bootstrap has the advantage to produce
confidence intervals for such statistics without theoretical probability distribution and finally we
accept the RE and CE for which the lower confidence margin at 95% are positive. This is more
constraining than just accepting all positive RE and CE. For additional verification, we also
present traditional split-sample procedure results that divided the full period into two subsets of
equal length (Meko and Graybill 1995).

To identify the extreme March–April cold and warm events in the reconstruction, standard
deviation (SD) values were used.  Years one and two SD above and below the mean were
identified as warm, very warm, cold, and very cold years, respectively. As a way to assess the
spatial representation of our temperature reconstruction, we conducted a spatial field correlation
analysis between reconstructed values and the gridded CRU TS3.23 temperature field (Jones and
Harris 2008) for a broad region of the Mediterranean over the entire instrumental period (ca.
1930–2002).   Finally, we compared our temperature reconstruction and also precipitation signal
(PC1) against existing gridded temperature and hydroclimate reconstructions for Europe over the
period 1800–2002. We performed spatial correlation analysis between [1] our temperature
reconstruction and gridded temperature reconstructions for Europe (Xoplaki et al. 2005,
Luterbacher et al. 2016) and OWDA (Cook et al. 2015); and [2] PC1 and summer precipitation
reconstruction (Pauling et al., 2006) and Old World Drought Atlas (OWDA) (Cook et al. 2015).
To assess the significance of the correlation between our reconstruction (y) and gridded
reconstructions (xj, j=1…N), we have calculated significance thresholds based on a Monte-Carlo
technique. For each gridpoint j, we have calculated the correlation between xj and y, but with a
random permutation of the values of our reconstruction. This is repeated 1000 times with a
different permutation. The 1000 correlation coefficients so obtained are expected to be zero as
the correlation is established on non corresponding years. The 95th quantile of these 1000
coefficients is assumed to be passed in less than 5% of the cases. Then a correlation coefficient
with a higher value is considered as positive with a 95% confidence. These thresholds are
obtained with a common permutation for all xj so that the spatial structure is conserved in the
tests. The sign + is assigned to the xj with a correlation higher than an expected value under the
non-correlation hypothesis.

**2  Results and Discussion**
2.1 Tree-ring chronologies
In addition to 15 chronologies developed by previous studies, we produced six *P. nigra*, one *P.*
*sylvestris*, one *A. nordmanniana / P. orientalis* chronologies for this study (Table 2). The Çorum
district produced two *P. nigra* chronologies: one the longest (KAR; 627 years long) and the other
the most sensitive to climate (SAH; mean sensitivity value of 0.25). Previous investigations of
climate-tree growth relationships reported a mean sensitivity range of 0.13–0.25 for *P. nigra* in
Turkey (Köse 2011, Akkemik et al. 2008). The KAR, SAH, and ERC chronologies (with mean
sensitivity values from 0.22 to 0.25) were classified as very sensitive, and the SAV, HCR, and
PAY chronologies (mean sensitivity values range 0.17–0.18) contained values characteristic of
being sensitive to climate. The lowest mean sensitivity value was obtained for the ART *A.*
*nordmanniana / P. orientalis* chronology. Nonetheless, this chronology retained a statistically
significant temperature signal ($p < 0.05$).

231       2.2 Tree-ring growth-climate relationship

RFA coefficients of May to August precipitation are positively correlated with most of the tree-
ring series (Fig. 2) and among them, May and June coefficients are generally significant. The
first principal component of the 23 chronologies, which explains 47% of the tree-growth
variance, is highly correlated with May–August total precipitation, statistically ($r = 0.65$, $p <$
0.001) and visually (Fig. 3). The high correlation was expected given that numerous studies also
found similar results in Turkey (Akkemik 2000a, Akkemik 2000b, Akkemik 2003, Akkemik et
al. 2005, Akkemik et al. 2008, Akkemik & Aras 2005, Hughes et al. 2001, D'Arrigo &  Cullen
2001, Touchan et al. 2003; Touchan et al. 2005a, Touchan et al. 2005b, Touchan et al. 2007,
Köse et al. 2011, Köse et al. 2012,  Köse et al. 2013, Martin-Benitto et al. 2016). The influence
of temperature was not as strong as May–August precipitation on radial growth, although
generally positive in early spring (March and April) (Fig. 2). Conversely, the ART chronology
from northeastern Turkey contained a strong temperature signal, which was significantly positive
in March.
Correlation maps representing influence of May-August precipitation (Fig. 4a) and March-April
temperature (Fig 4b) also showed that strength of the summer precipitation signal is higher and
significant almost all over the Turkey. Higher precipitation in summer has a positive effect on
tree-growth, because of long-lasting dry and warm conditions over the Turkey (Türkeş 1996b,
Köse et al. 2012). Spring precipitation signal are generally positive and significant only for four
tree-ring sites. The sites located at the upper distributions of the species are generally showed
higher correlations. The highest correlations obtained for *Picea/Abies* chronology (ART) from
the Caucasus, and for *Pinus nigra* chronology (HCR) from the upper (about 1900 m) and
southeastern distribution of the species. This black pine forest was still partly covered by snow
from previous year during the field work in fall. Higher temperatures in spring maybe cause
snow melt earlier and lead to produce larger annual rings. In addition to these chronologies, we
also used the chronologies that revealed the influence of precipitation, as well as temperature to
reconstruct March–April temperature.


2.3 March-April temperature reconstruction
The higher order PCs of the 23 chronologies are significantly correlated with the March–April
temperature and, by nature, are independent on the precipitation signal (Table 3). The best
selection for fit temperature are obtained with the $PC_3$, $PC_4$, $PC_5$, $PC_7$, $PC_9$, $PC_{10}$, $PC_{15}$, $PC_{17}$,
$PC_{21}$, which explains together 25% of the tree-ring chronologies. So the temperature signal
remains important in the tree-ring chronologies and can be reconstructed. The advantage to
separate both signals through orthogonal PCs enable to remove an unwanted noise for our
temperature reconstruction. Thus, $PC_1$ was not used as potential predictor of temperature because
it is largely dominated by precipitation (Table 3, Fig. 3). The last two PCs contain a too small
part of the total variance to be used in the regressions. However, even if Jolliffe (1982) and Hadi
& Ling (1998) claimed that certain PCs with small eigenvalues (even the last one), which are
commonly ignored by principal components regression methodology, may be related to the
independent variable, we must be cautious with that because they may be much more dominated
by noise than the first ones. So, the contribution of each PC to the regression sum of squares is
also important for selection of PCs (Hadi & Ling 1998). The findings of Jolliffe (1982) and Hadi
& Ling (1998) provide a justification for using non-primary PCs, (*e.g.*, of second and higher
order) in our regression, given that correlations with temperature may be over-powered by
affects from precipitation in our study area (Cook 2011, personal communication).

Using this method, the calibration and verification statistics indicated a statistically significant
reconstruction (Table 4, Fig. 5). For additional verification, we also present split-sample
procedure results. Similarly bootstrap results, the derived calibration and verification tests using
this method indicated a statistically significant RE and CE values (Table 5).

The regression model accounted for 67% (Adj. $R^2 = 0.64$, $p < 0.0001$) of the actual temperature
variance over the calibration period (1930–2002). Also, actual and reconstructed March–April
temperature values had nearly identical trends during the period 1930–2002 (Fig. 5). Moreover,
the tree-ring chronologies successfully simulated both high frequency and warming trends in the
temperature data during this period. The reconstruction was more powerful at classifying warm
events rather than cold events. Over the last 73 years, eight of ten warm events in the
instrumental data were also observed in the reconstruction, while five of nine cold events were
captured. Similarly, previous tree-ring based precipitation reconstructions for Turkey (Köse et al.
2011; Akkemik et al. 2008) were generally more successful in capturing dry years rather than
wet years.

Our temperature reconstruction on the 1800–2002 period is obtained by bootstrap regression,
using 1000 iterations (Fig. 6). The confidence intervals are obtained from the range between the
$2.5^{th}$ and the $97.5^{th}$ percentiles of the 1000 simulations. Low frequency variability of our spring
temperature reconstruction showed larger variability in nineteenth century than twentieth
century. For the pre-instrumental period (1800–1929), a total of 23 cold (1813, 1818, 1821,
1824, 1837, 1848, 1854, 1858, 1860, 1869, 1877–1878, 1880–1881, 1883, 1897–1898, 1905–
1907, 1911–1912, 1923) and 13 warm (1801–1802, 1807, 1845, 1853, 1866, 1872–1873, 1879,
1885, 1890, 1901, 1926) events were determined. After comparing our results with event years
obtained from May–June precipitation reconstructions from western Anatolia (Köse et al. 2011),
the cold years 1818, 1848, and 1897 appeared to coincide with wet years and 1881 was a very
wet year for the entire region. Furthermore, these years can be described as cold (in March–
April) and wet (in May–June) for western Anatolia.

Among the warm periods in our reconstruction, conditions during the year 1879 were dry, 1895
wet, and 1901 very wet across the broad region of western Anatolia (Köse et al. 2011). Hence,
we defined 1879 as a warm (in March–April) and dry year (in May–June), and 1895 and 1901
were warm and wet years. In the years 1895 and 1901 the combination of a warm early spring
and a wet late spring-summer caused enhanced radial growth in Turkey, interpreted as longer
growing seasons without drought stress.

Of these event years, 1897 and 1898 were exceptionally cold and 1845, 1872 and 1873 were
exceptionally warm. During the last 200 years, our reconstruction suggests that the coldest year
was 1898 and the warmest year was 1873. The reconstructed extreme events also coincided with
accounts from historical records. Server (2008) recounted the winter of 1898 as characterized by
anomalously cold temperatures that persisted late into the spring season. A family, who brought
their livestock herds up into the plateau region in Kırşehir seeking food and water were suddenly
covered in snow on 11 March 1898. This account of a late spring freeze supports the
reconstruction record of spring temperatures across Turkey, and offers corroboration to the
quality of the reconstructed values.

Seyf (1985) reported that extreme summer temperature during the year 1873 resulted in
widespread crop failure and famine. Historical documents recorded an infamous drought-derived
famine that occurred in Anatolia from 1873 to 1874 (Quataert, 1996, Kuniholm, 1990), which
claimed the lives of 250,000 people and a large number of cattle and sheep (Faroqhi, 2009). This
drought caused widespread mortality of livestock and depopulation of rural areas through human
mortality, and migration of people from rural to urban areas. Further, the German traveler
Naumann (1893) reported a very dry and hot summer in Turkey during the year 1873 (Heinrich
et al, 2013). Conditions worsened when the international stock exchanges crashed in 1873
(Zürcher, 2004). Our temperature record suggests that dry conditions during the early 1870s
were possibly exacerbated by warm spring temperatures that likely carried into summer. A
similar pattern of intensified drought by warm temperatures was demonstrated recently by
Griffin and Anchukaitis (2014) for the current drought in California, USA.

Extreme cold and warm events were usually one year long, and the longest extreme cold and

warm events were two and three years, respectively. These results were similar with durations of

extreme wet and dry events in Turkey (Touchan et al. 2003, Touchan et al. 2005a, Touchan et al.

2005b, Touchan et al. 2007, Akkemik & Aras, 2005, Akkemik et al. 2005, Akkemik et al. 2008,

Köse et al. 2011, Güner et al. 2016). Moreover, seemingly innocuous short-term warm events,

such as the 1807 event, were recorded across the Mediterranean and in high elevations of the

European regions. Casty et al. (2005) reported the year 1807 as being one of the warmest alpine

summers in the European Alps over the last 500 years. As such, a drought record from Nicault et

al. (2008) echoes this finding, as a broad region of the Mediterranean basin experienced drought

conditions.

348

349

350

Heinrich et al. (2013) analyzed winter-to-spring (January–May) air temperature variability in

Turkey since AD 1125 as revealed from a robust tree-ring carbon isotope record from *Juniperus*

*excelsa*. Although they offered a long-term perspective of temperature over Turkey, the

reconstruction model, which covered the period 1949–2006, explained 27% of the variance in

temperature since the year 1949. In this study, we provided a short-term perspective of

temperature fluctuation based on a robust model (calibrated and verified 1930–2002; Adj. $R^2 =$

0.64; $p < 0.0001$). Yet, the Heinrich et al. (2013) temperature record did not capture the 20[th]

century warming trend as found elsewhere (Wahl et al. 2010). However, their temperature trend

does agree with trend analyses conducted on meteorological data from Turkey and other areas in

the eastern Mediterranean region. The warming trend seen during our reconstruction calibration
period (1930–2002) was similar to the data shown by Wahl et al. (2010) across the region and
hemisphere. Further, the warming trends seen in our record agrees with data presented by Turkes
& Sumer (2004), of which they attributed to increased urbanization in Turkey. Considering long-
term changes in spring temperatures, the 19$^{th}$ century was characterized by more high-frequency
fluctuations compared to the 20$^{th}$ century, which was defined by more gradual changes and
includes the beginning of decreased DTRs in the region (Turkes & Sumer 2004).

2.4 Comparison with instrumental gridded data and spatial reconstructions

Spatial correlation analysis revealed that our network-based temperature reconstruction was
representative of conditions across Turkey, as well as the broader Mediterranean region (Fig. 7).
During the period 1930–2002, estimated temperature values were highly significant ($r$ range 0.5–
0.6, $p < 0.01$) with instrumental conditions recorded from southern Ukraine to the west across
Romania, and from northern areas of Libya and Egypt to the east across Iraq. The strength of the
reconstruction model is evident in the broad spatial implications demonstrated by the
temperature record. Thus, we interpret warm and cold periods and extreme events within the
record with high confidence.

We compared our tree-ring based temperature reconstruction with existing gridded temperature
reconstructions for Europe (Xoplaki et al. 2005, Luterbacher et al. 2016) and the Old World
Drought Atlas (OWDA) (Cook et al. 2015) for further validation of the reconstruction (Fig. 8a,
b, c, respectively). Spatial correlations over the past 200 years were lower with reconstructed
European summer temperature (May to July) (Fig. 8b). Yet, we expected this result because of
the paucity of Turkey-derived proxies in the other reconstructions, as well as the differing
seasons involved across the reconstructions. Similarly, our reconstruction showed weak
correlations with summer drought index over Turkey. Beside comparing different seasons,
perhaps this is because less precipitation begets drought conditions rather than high temperature
in the region. The highest and significant ($p < 0.05$) correlations were found with European
spring (March to May) temperature reconstruction over southeastern Europe, which are stronger
over Turkey (Fig. 8a). We used the mean of corresponding grid points from European spring
temperature reconstruction over the study area (36–42º N, 26–38º E) to show how the correlation
changed over time (Fig. 9). The correlation coefficient was highly significant ($r = 0.76$, $p <$
0.001) during our calibration period (1930–2002). We found lower but still significant
correlation ($r = 0.35$, $p < 0.10$) for the period of 1901–1929, which climatic records are very few
over the region while available data has sufficient quality for most part of Europe. These results
give additional verification for our reconstruction. Moreover, our reconstruction has a weak,
insignificant relationship ($r = 0.13$, $p > 0.10$) during the 19[th] century. This may be related to poor
reconstructive skill of European spring temperature reconstruction over Turkey, which contains
few proxies from the country (Xoplaki et al. 2005, Luterbacher et al. 2004). Nonetheless, these
results demonstrate that tree-ring chronologies from Turkey can serve as useful temperature
proxies for further spatial temperature reconstructions to fill the gaps in the area.

We also compared the precipitation signal (PC1) obtained from our tree-ring network with Old
World Drought Atlas (OWDA) (Cook et al. 2015) and gridded European summer precipitation
reconstruction (Pauling et al., 2006) to test the strength of the signal spatially (Fig. 8d and e,
respectively). We calculated highly significant positive correlations with summer drought index
over Turkey and neighboring European countries such as Greece, Bulgaria, and Romania, Italy
while significant correlations are lower for the northern Mediterranean countries (Fig. 8d). These
results showed that summer precipitation signal represented by PC1 is very strong not only on
instrumental period, but also on pre-instrumental period, and represents a large spatial coverage.
We found low and insignificant correlations over Turkey and Mediterranean countries with
European summer precipitation reconstruction (Fig. 8e). Pauling et al. (2006) stated that poor
reconstructive skills determined over Turkey because of few instrumental record before
the1930s.

**4 Conclusions**

In this study, we used a broad network of tree-ring chronologies to provide the first tree-ring
based temperature reconstruction for Turkey and identified extreme cold and warm events during
the period 1800–1929 CE. Similar to the precipitation reconstructions against which we compare
our air temperature record, extreme cold and warm years were generally short in duration (one
year) and rarely exceeded two-three years in duration. The coldest and warmest years over
western Anatolia were experienced during the 19th century, and the 20th century is marked by a
temperature increase.

Reconstructed temperatures for the 19th century suggest that more short-term fluctuations
occurred compared to the 20th century. The gradual warming trend shown by our reconstruction
calibration period (1930–2002) is coeval with decreases in spring DTRs. Given the results of
Turkes and Sumer (2004), the variations in short- and long-term temperature changes between
the 19[th] and 20[th] centuries might be related to increased urbanization in Turkey.

We highlight that the 20[th] century warming trend is unprecedented within the context of the past
*ca.* 200 years, especially over the past *ca.* 15 years. Correlations with gridded climate fields and
other climate reconstructions from the region revealed that our network-based temperature
reconstruction was representative of conditions across Turkey, as well as the broader
Mediterranean region. Expanding the tree-ring network across Turkey, especially to the east, will
improve the spatial implications of future temperature reconstructions.

The study revealed the potential for reconstructing temperature in an area previously thought
impossible, especially given the strong precipitation signals displayed by most tree species
growing in the dry Mediterranean climate that characterizes broad areas of Turkey. Our
reconstruction only spans 205 years due to the shortness of the common interval for the
chronologies used in this study, but the possibility exists to extend our temperature
reconstruction further back in time by increasing the sample depth with more temperature-
sensitive trees, especially from northeastern Turkey. Thus future research will focus on
increasing the number of tree-ring sites across Turkey, and maximizing chronology length at
existing sites that would ultimately extend the reconstruction back in time.




**Acknowledgements**

**Acknowledgements**

This research was supported by The Scientific and Technical Research Council of Turkey
(TUBITAK); Projects ÇAYDAG 107Y267 and YDABAG 102Y063. N. Köse was supported by
The Council of Higher Education of Turkey. We are grateful to the Turkish Forest Service
personnel and Ali Kaya, Umut Ç. Kahraman and Hüseyin Yurtseven for their invaluable support
during our field studies. We thank to Dr. Ufuk Turuncoğlu for his help on spatial analysis.  J.
Guiot was supported by the Labex OT-Med (ANR-11-LABEX-0061), French National Research
Agency (ANR).

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

Table 1. Site information for the new chronologies developed by this study in Turkey.

| Site name | Site code | Species | No. trees/ cores | Aspect | Elev. (m) | Lat. (N) | Long. (E) |
|---|---|---|---|---|---|---|---|
| Çorum, Kargı, Karakise kayalıkları | KAR | *Pinus nigra* | 22 / 38 | SW | 1522 | 41°11' | 34°28' |
| Çorum, Kargı, Şahinkayası mevkii | SAH | *P. nigra* | 12 / 21 | S | 1300 | 41°13' | 34°47' |
| Bilecik, Muratdere | ERC | *P. nigra* | 12 / 25 | SE | 1240 | 39°53' | 29°50' |
| Bolu, Yedigöller, Ayıkaya mevkii | BOL | *P. sylvestris* | 10 / 20 | SW | 1702 | 40°53' | 31°40' |
| Eskişehir, Mihalıççık, Savaş alanı mevkii | SAV | *P. nigra* | 10 / 18 | S | 1558 | 39°57' | 31°12' |
| Kayseri, Aladağlar milli parkı, Hacer ormanı | HCR | *P. nigra* | 18 / 33 | S | 1884 | 37°49' | 35°17' |
| Kahramanmaraş, Göksun, Payanburnu mevkii | PAY | *P. nigra* | 10 / 17 | S | 1367 | 37°52' | 36°21' |
| Artvin, Borçka, Balcı işletmesi | ART | *Abies nordmanniana Picea orientalis* | 23 / 45 | N | 1200–2100 | 41°18' | 41°54' |


Table 2. Summary statistics for the new chronologies developed by this study in Turkey.

| | Total chronology | | | Common interval | | |
|---|---|---|---|---|---|---|
| Site Code | Time span | 1st year (*EPS > 0.85) | Mean sensitivity | Time span | Mean correlations: among radii /between radii and mean | Variance explained by PC1 (%) |
| KAR | 1307–2003 | 1620 | 0.22 | 1740–1994 | 0.38 / 0.63 | 41 |
| SAH | 1663–2003 | 1738 | 0.25 | 1799–2000 | 0.42 / 0.67 | 45 |
| ERC | 1721–2008 | 1721 | 0.23 | 1837–2008 | 0.45 / 0.69 | 48 |
| BOL | 1752–2009 | 1801 | 0.18 | 1839–1994 | 0.32 / 0.60 | 36 |
| SAV | 1630–2005 | 1700 | 0.17 | 1775–2000 | 0.33 / 0.60 | 38 |
| HCR | 1532–2010 | 1704 | 0.18 | 1730–2010 | 0.38 / 0.63 | 40 |
| PAY | 1537–2010 | 1790 | 0.18 | 1880–2010 | 0.28 / 0.56 | 32 |
| ART | 1498–2007 | 1624 | 0.12 | 1739–1996 | 0.37 / 0.60 | 41 |

*EPS = Expressed Population Signal [Wigley et al., 1984]

Table 3. Principal components analysis statistics for the Turkey temperature reconstruction model.

| | Explained variance | Correlation coefficients with | | The chronologies represented by higher magnitudes** in the eigenvectors |
|---|---|---|---|---|
| | | May–August PPT | March–April TMP | |
| | (%) | | | |
| PC1 | 46.57 | 0.65 | 0.19 | KAR, KIZ, TEF, BON,USA,TUR, CAT, INC, ERC, YAU, SAV, TAN, SIU |
| PC2 | 7.86 | –0.07 | 0.15 | KAR, SAV, TIR, BOL, YAU, ESK, TEF,BON, SIU |
| PC3* | 4.93 | 0.04 | –0.48 | HCR, PAY, BOL, YAU, SIA |
| PC4* | 4.68 | 0.11 | 0.17 | TEF, KEL, FIR, SIA, KIZ, SIU, ART |
| PC5* | 4.42 | –0.25 | 0.27 | SAH, TIR, FIR, ART |
| PC6 | 3.73 | 0.15 | –0.14 | KIZ, FIR, SAV, KAR, TIR, PAY, ESK, TEF, BON, ART |
| PC7* | 3.56 | 0.19 | 0.18 | KIZ, BON, BOL, YAU, HCR, PAY, INC |
| PC8 | 2.87 | 0.26 | 0.01 | HCR, ESK, BON, FIR, ERC, SIA |
| PC9* | 2.45 | 0.16 | 0.17 | PAY, USA, BOL, YAU, TIR, HCR, FIR, SIA, SIU |
| PC10* | 2.21 | 0.14 | –0.08 | TUR, CAT, SAV, SIA, KEL, ERC, SIU |
| PC11 | 2.09 | –0.36 | –0.20 | HCR, TEF, USA, INC, PAY, TUR, SAV, SIU |
| PC12 | 1.80 | –0.12 | 0.05 | TEF, CAT, YAU HCR, ESK, USA, BOL, SIA |
| PC13 | 1.63 | –0.06 | 0.17 | TEF, TUR, BOL, KAR, YAU, SIA |
| PC14 | 1.55 | –0.14 | 0.06 | TIR, USA, FIR, TUR, YAU, KAR, BON |
| PC15* | 1.50 | –0.20 | –0.14 | KIZ, BON, USA, ESK, INC, BOL |
| PC16 | 1.31 | 0.04 | 0.08 | SAH, HCR, INC, YAU, SAV, KAR, FIR, BOL, SIU |
| PC17* | 1.25 | 0.15 | 0.19 | SAH, SIU, KAR, ESK, TUR, ERC |
| PC18 | 1.14 | 0.13 | 0.02 | KAR, TEF, TUR, SAV, BON, CAT |
| PC19 | 1.09 | 0.16 | –0.11 | PAY, INC, SAV, HCR, KEL, CAT, TAN |
| PC20 | 0.95 | –0.15 | –0.01 | TIR, SAH, CAT |
| PC21* | 0.89 | 0.06 | –0.28 | TUR, INC, TIR, SAV |
| PC22 | 0.85 | 0.44 | 0.10 | KIZ, SAH, BON, YAU, SIU |
| PC23 | 0.67 | –0.22 | –0.02 | TAN, KEL, TUR, CAT |

"*" indicates the PCs, which used in the reconstruction as predictors

"**" which exceed ±0.2 value.



Table 4. Calibration and verification statistics of bootstrap method (1000 iterations
applied) showing the mean values based on the 95% confidence interval (CI).

|  |  | Mean (95% CI) |
| --- | --- | --- |
| Calibration | RMSE | 0.65 (0.52; 0.77) |
|  | $R^2$ | 0.73 (0.60; 0.83) |
| Verification | RE | 0.54 (0.15; 0.74) |
|  | CE | 0.51 (0.04; 0.72) |
|  | RMSEP | 0.88 (0.67; 1.09) |

*RMSE* root mean squared error; $R^2$ coefficient of determination; *RE* reduction of error; *CE*
coefficient of efficiency; *RMSEP* root mean squared error prediction

Table 5. Calibration and cross-validation statistics for the Turkey temperature reconstruction
model.

| Calibration Period | Verification Period | Adj. $R^2$ | F | RE | CE |
| --- | --- | --- | --- | --- | --- |
| 1930–1966 | 1967–2002 | 0.55 | 5.91 | 0.64 | 0.58 |
|  |  |  | $p < 0.0001$ |  |  |
| 1967–2002 | 1930–1966 | 0.71 | 10.45 | 0.63 | 0.46 |
|  |  |  | $p < 0.0001$ |  |  |




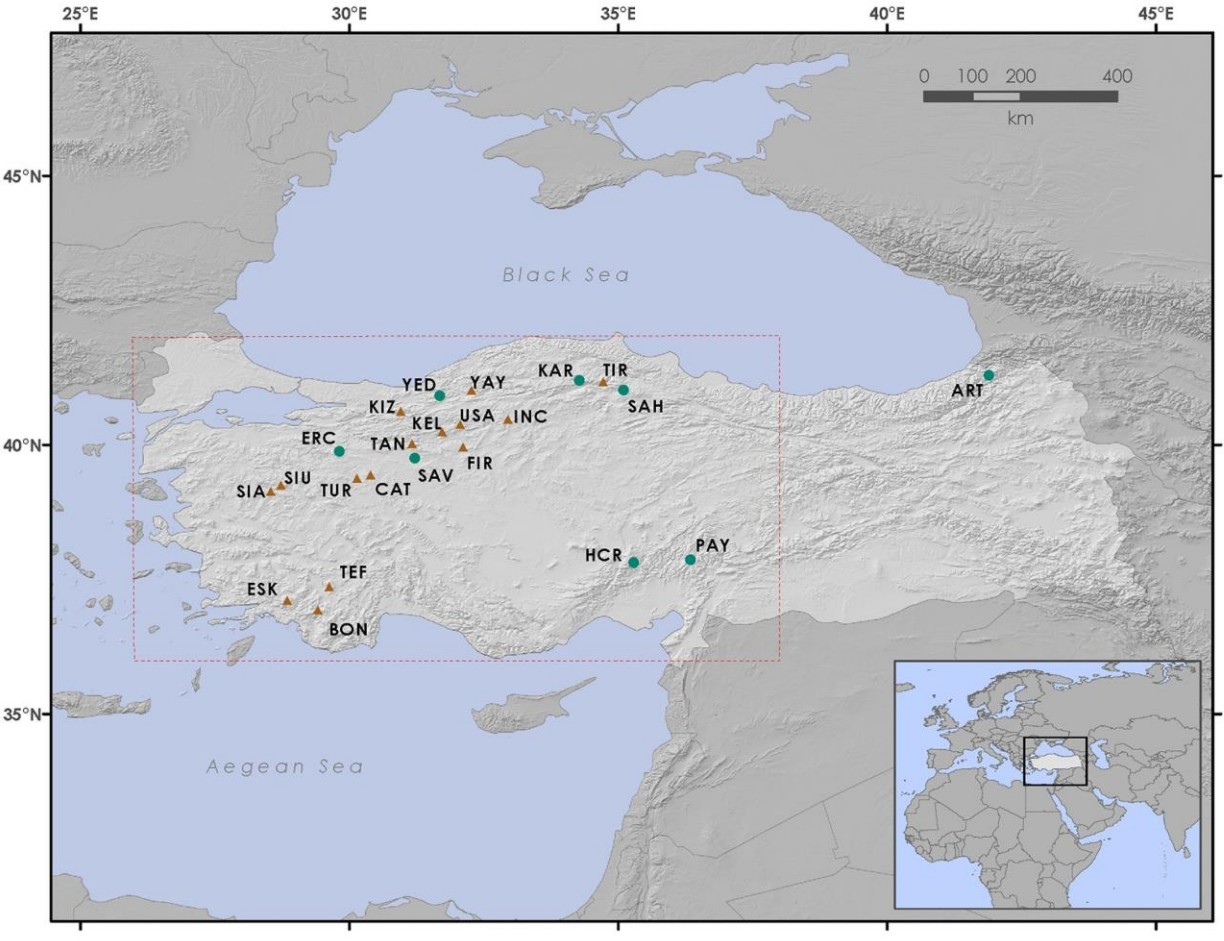

**Figure 1.** Tree-ring chronology sites in Turkey used to reconstruct temperature. Circles
represent the new sampling efforts from this study and the triangles represent previously-
published chronologies (YAY, SIA, SIU: Mutlu et al. 2011; TIR: Akkemik et al. 2008; TAN:
Köse et al. unpublished data; KIZ, ESK, TEF, BON, KEL, USA, FIR, TUR: Köse et al. 2011;
CAT, INC: Köse et al. 2005). The box (dashed line) represents the area for which the
temperature reconstruction was performed.



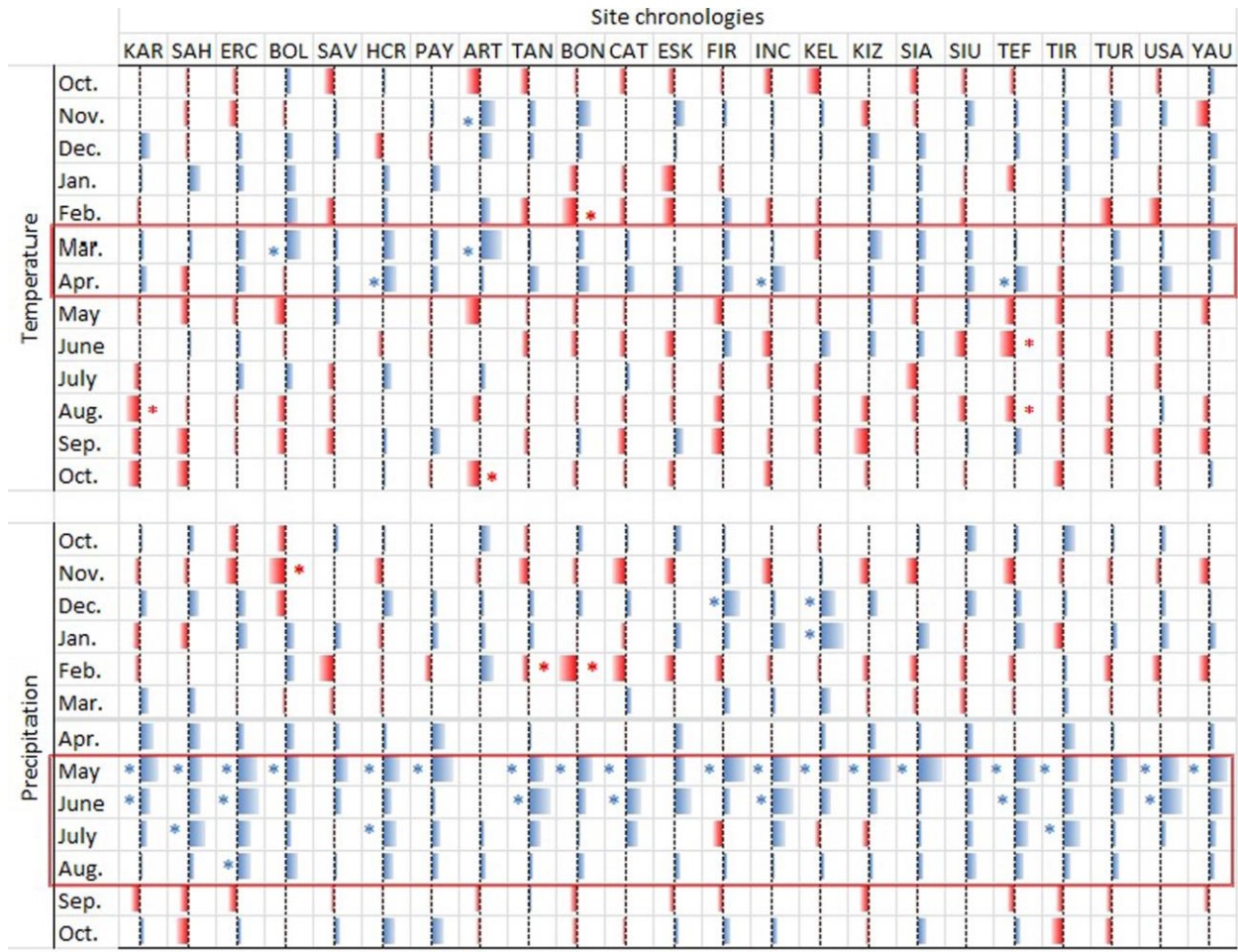

**Figure 2.** Summary of response function results of 23 chronologies. Red color represents negative effects of climate variability on tree ring width; blue color represents positive effects of climate variability on tree ring width. "*" indicates statistically significant response function confidents ($p < 0.05$). Each response function includes 13 weights for average monthly temperatures and 13 monthly precipitations from October of the prior year to October of current year.

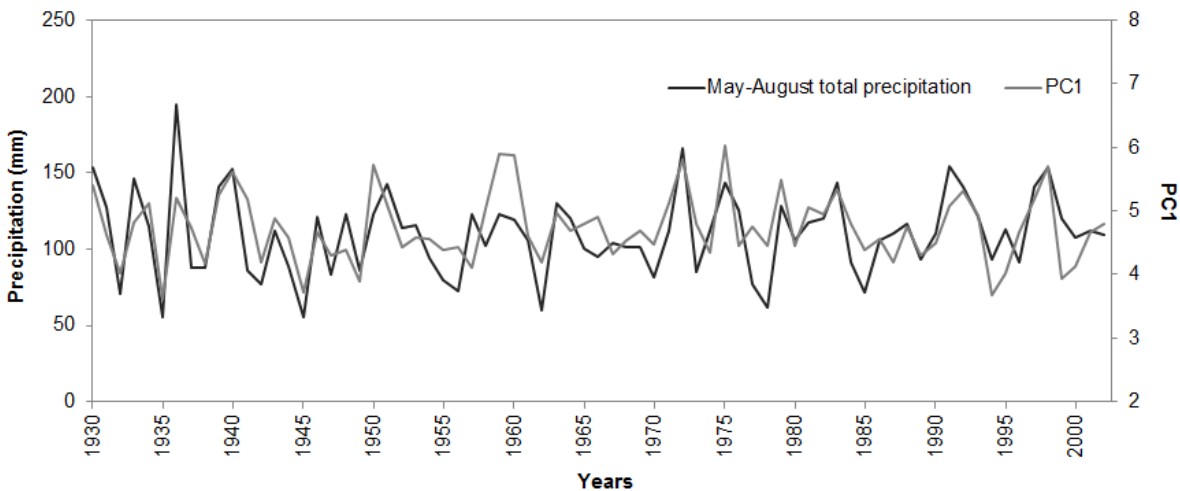


**Figure 3.** The comparison of May–August total precipitation (black) and the first principal
component of 23 tree-ring chronologies (gray). Correlation coefficient between two time series is
0.65 ($p < 0.001$).



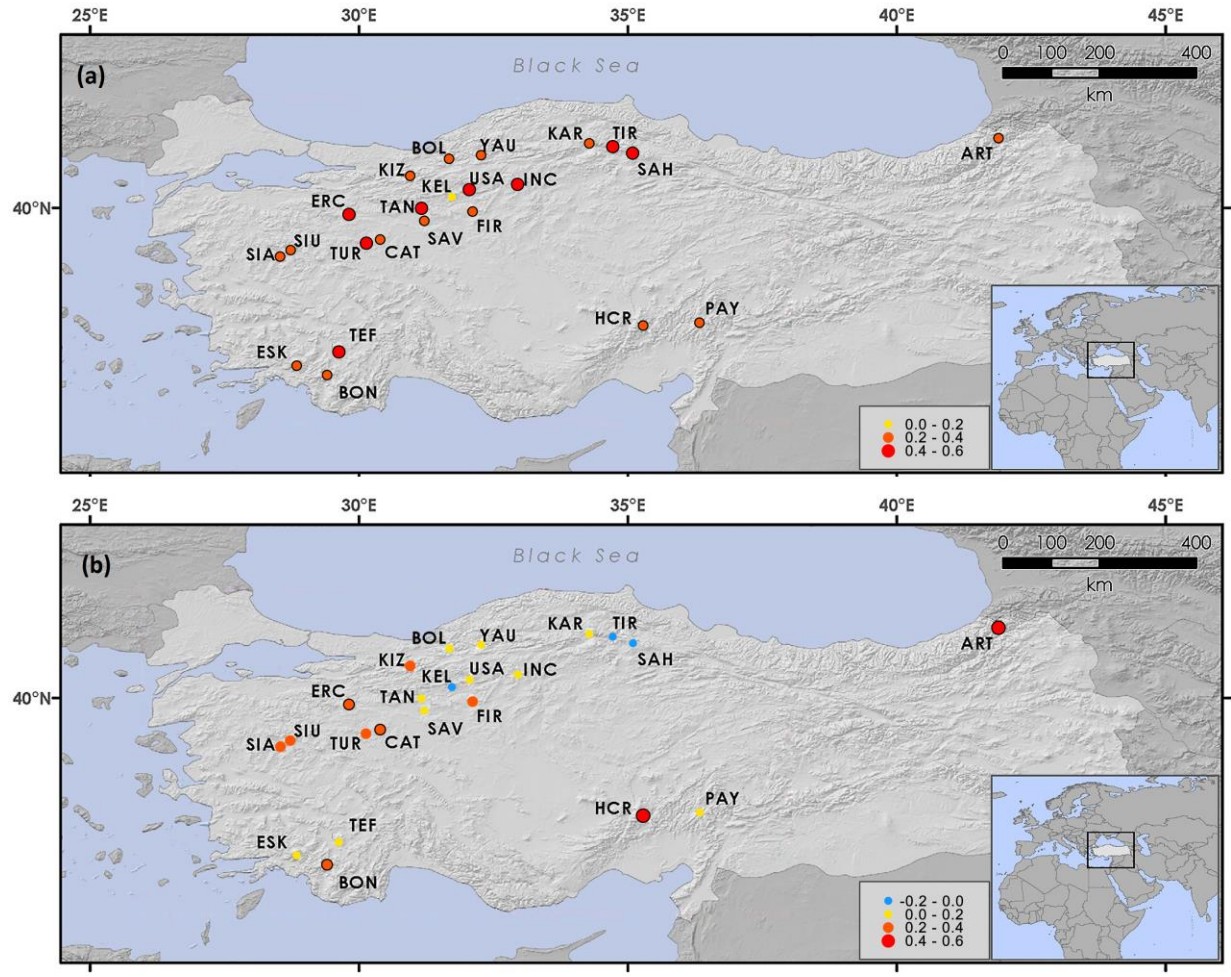

**Figure 4.** Maps showing Pearson's correlation coefficients between the sites chronologies and (a) May–August total precipitation and (b) March-April mean temperature for the period 1930–2012. For each site, the closest gridded (0.5° x 0.5°) climate data obtained from CRU dataset were used. Graduated circle size and color correspond to correlation coefficient versus the climate variable. Black lines surrounding circles represent significant correlation coefficients ($p < 0.05$).


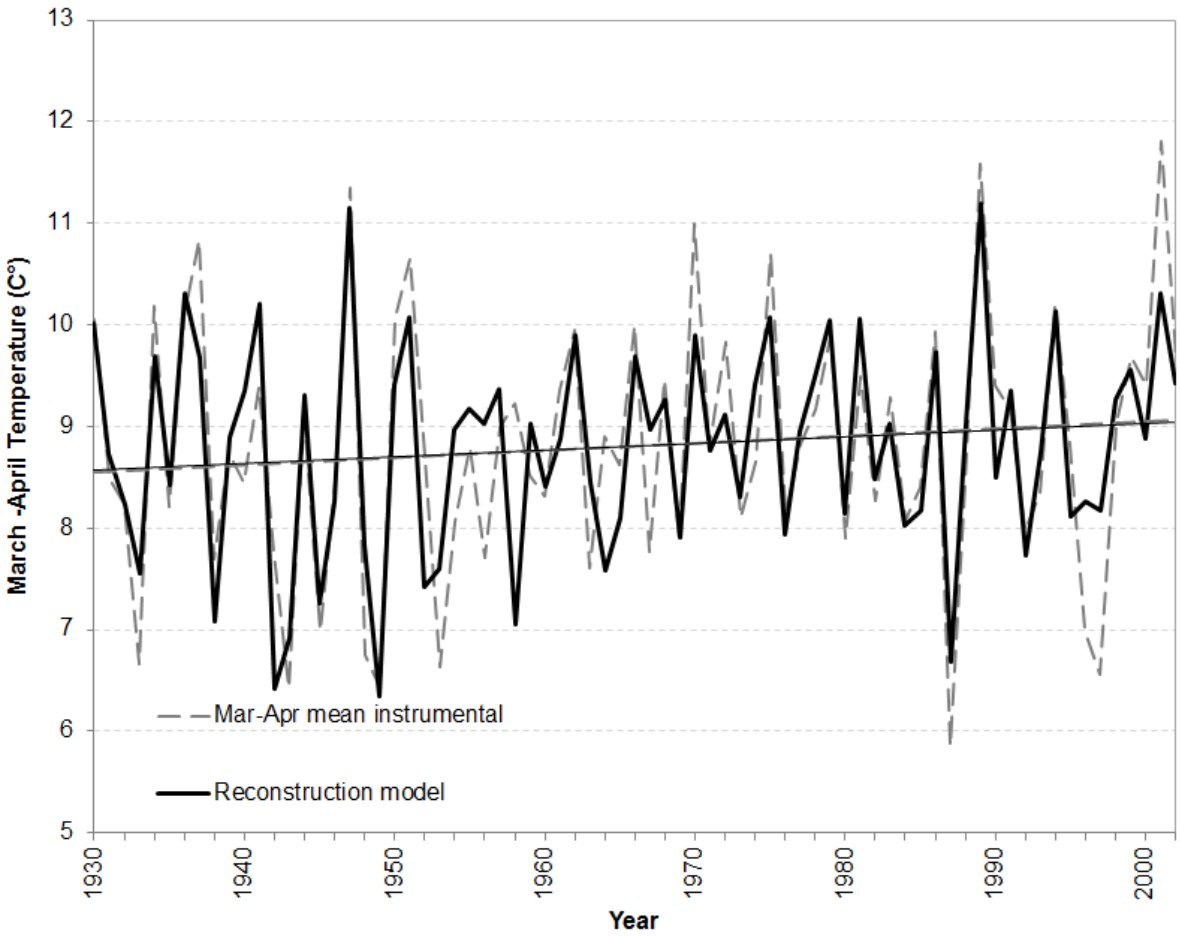


**Figure 5.** Actual (instrumental) and reconstructed March–April temperature (°C). Dashed lines
(dark grey) represent actual values and solid lines (black) represent reconstructed values shown
with trend lines (linear dashed grey and linear black lines, respectively). The tendency to warm
up at the reconstructed temperature is in good agreement with the trend in instrumental data.








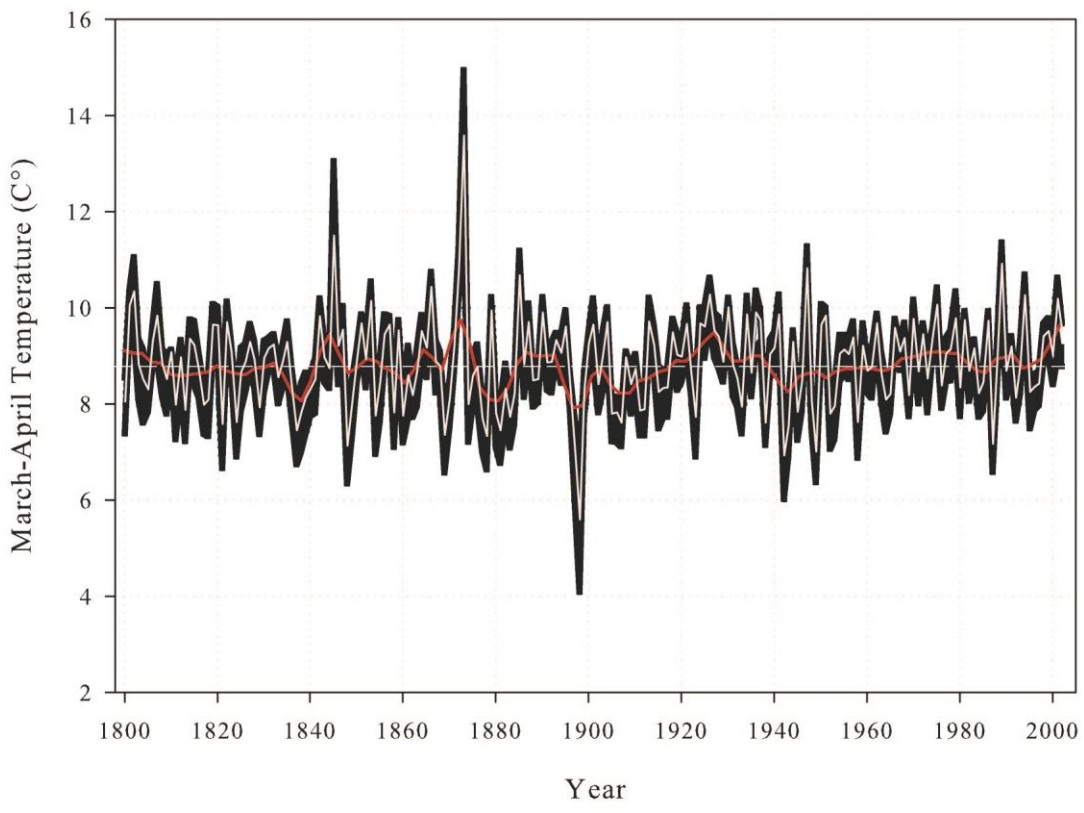


**Figure 6.** March–April temperature reconstruction for Turkey for the period 1800–2002

CE. The central horizontal line (dashed white) shows the reconstructed long-term mean and

does not include instrumental data; black background denotes Monte Carlo ($n = 1000$)

bootstrapped 95% confidence limits; and the red line shows 13-year low-pass filter values.






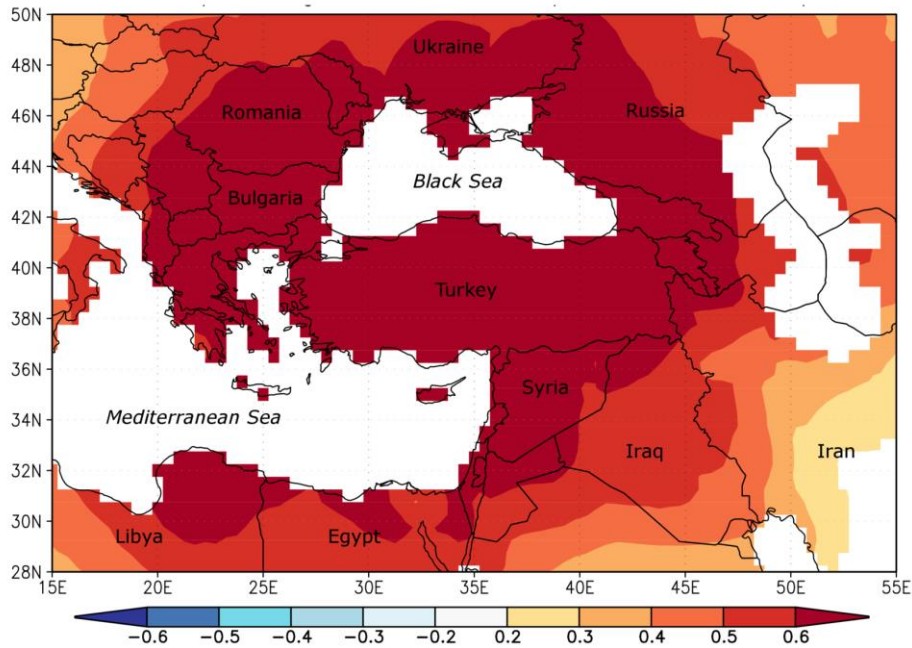


**Figure 7.** Spatial correlation map for the March–April temperature reconstruction. Spatial

field correlation map showing statistical relationship between the temperature

reconstruction and the gridded temperature field at 0.5º intervals (CRU TS3.23; Jones and

Harris 2008) during the period 1930–2002 over the Mediterranean region. For each grid,

calculated correlation coefficient from 0.20 to 0.60 is significant ($p < 0.05$).

729

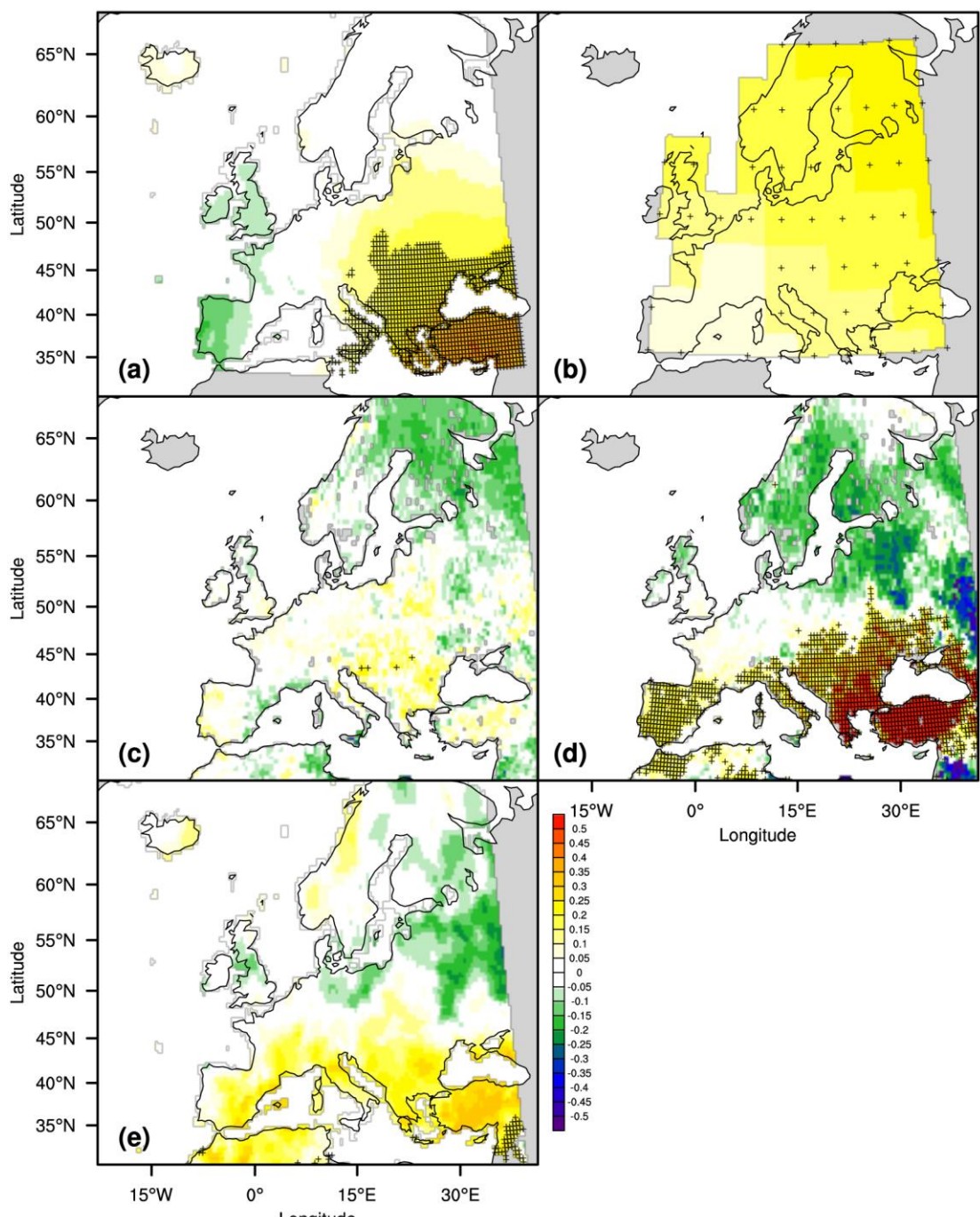

**Figure 8.** Spatial correlation maps for the March–April temperature reconstruction and

precipitation signal (PC1) obtained from tree-ring data set during the period 1800–2002 over

Europe. Maps demonstrate spatial field correlations between our temperature reconstruction and

(a) gridded spring temperature reconstruction for Europe (Xoplaki et al. 2005), (b) gridded

summer temperature reconstruction for Europe (Luterbacher et al. 2016), (c) Old World Drought
Atlas (OWDA; Cook et al. 2015). Panels (d) and (e) show spatial correlations between PC1 and
OWDA (Cook et al. 2015) and gridded European summer precipitation reconstruction (Pauling
et al., 2006), respectively. '+' represents significant correlation coefficients ($p < 0.05$).

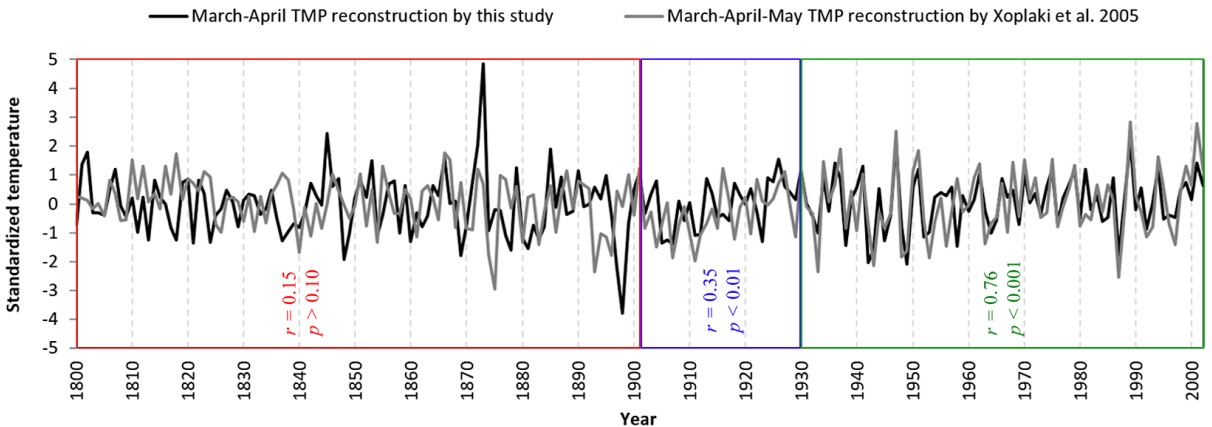


**Figure 9.** Comparison of March-April temperature reconstruction (gray) with the mean of
corresponding grid points from European spring (March to May) temperature
reconstruction (Xoplaki et al. 2005; black) over the study area (36–42º N, 26–38º E). The
indicated correlation coefficients are calculated for instrumental period (also calibration
period for this study) (1930–2002; $r = 0.76$, $p < 0.001$); for the pre-instrumental period of
Turkey, while instrumental data has sufficient quality for most part of Europe (1901–1929;
$r = 0.35$, $p < 0.10$); and for pre-instrumental period (1800–1900; $r = 0.13$, $p < 0.10$).

**Spring temperature variability over Turkey since 1800 CE reconstructed**

**from a broad network of tree-ring data**

**Nesibe Köse[1],\*, H. Tuncay Güner[1], Grant L. Harley[2], Joel Guiot[3]**

[1]Istanbul University, Faculty of Forestry, Forest Botany Department 34473 Bahçeköy-Istanbul, Turkey
[2]University of Southern Mississippi, Department of Geography and Geology, 118 College Drive Box 5051, Hattiesburg, Mississippi, 39406, USA
[3] Aix-Marseille Université, CNRS, IRD, CEREGE UM34, ECCOREV, 13545 Aix-en-Provence, France

\*Corresponding author.  Fax: +90 212 226 11 13
 E-mail address: nesibe@istanbul.edu.tr

**Abstract**
The meteorological observational period in Turkey, which starts *ca.* 1930 CE, is too short for
understanding long-term climatic variability. Tree rings have been used intensively as proxy
records to understand summer precipitation history of the region, primarily because of having a
dominant precipitation signal. Yet,, the historical context of temperature variability is unclear.
Here we used higher order principle components of a network of 23 tree-ring chronologies to
provide a high-resolution spring (March–April) temperature reconstruction over Turkey during
the period 1800–2002. The reconstruction model accounted for 67% (Adj. $R^2 = 0.64$, $p < 0.0001$)
of the instrumental temperature variance over the full calibration period (1930–2002). The
reconstruction is punctuated by a temperature increase during the 20[th] century; yet extreme cold
and warm events during the 19[th] century seem to eclipse conditions during the 20[th] century..  We
found significant correlations between our March–April spring temperature reconstruction and
existing gridded spring temperature reconstructions for Europe over Turkey and southeastern
Europe. Moreover, the precipitation signal obtained from the tree-ring network (first principle
component) showed highly significant correlations with gridded summer drought index
reconstruction over Turkey and Mediterranean countries. Our results showed that, beside the
dominant precipitation signal, a temperature signal can be extracted from tree-ring series and
they can be useful proxies to reconstruct past temperature variability.
KEYWORDS: Dendroclimatology, Climate reconstruction, *Pinus nigra,* Principle component
analysis, Spring temperature.

## 1 Introduction

Long term meteorological observations in the Mediterranean region allow access to 100 years of instrumental reordings of temperature, precipitation and pressure in most of the region. Moreover, natural archives as well as documentary information provide resources with which to make sensitive climate reconstructions. An extensive body of literature details climate changes in the Mediterranean region over the last two millennia (c.f. Lionello, P. (Ed.)Luterbacher et al., 2012). Paleolimnological studies provide evidence that the Medieval Climatic Anomaly (MCA; 900–1300 CE) characterized warm and dry conditions over the Iberian Peninsula, while the Little Ice Age (LIA; 1300–1850 CE) brought opposite climate conditions, forced by interactions between the East Atlantic and North Atlantic Oscillation (Sanchez-Lopez et al. 2016). In addition, Roberts et al. (2012) highlighted an intriguing spatial dipole NAO pattern between the western and eastern Mediterranean region, which brought anti-phased warm (cool) and wet (dry) conditions during the MCA and LIA. The hydro-climate patterns revealed by previous investigations appear to have been forced not only by NAO, but other climate modes with non-stationary teleconnections across the region (Roberts et al. 2012).

The climate of Turkey is mainly characterized by Mediterranean macro climate (Türkeş, 1996a). Contrary to the most countries in the Mediterranean region, Turkey has relatively short meteorological records, which start in the 1930s, for understanding long-term climatic variability. On the other hand, proxy records such as speleothems (Fleitmann et al. 2009, Jex et al. 2010, Göktürk et al. 2011), lake sediments (Wick et al. 2003, Jones et al. 2006, Roberts et al. 2008, 2012, Kuzucuoğlu et al.2011, Woodbridge and Roberts 2011, Ülgen et al. 2012, Dean et

al. 2013) and tree-rings, have been used to reconstruct long term hydroclimate conditions over Turkey. Tree rings in particular have shown to provide useful information about the past climate of Turkey and were used intensively during the last decade to reconstruct precipitation in the Aegean (Griggs et al. 2007), Black Sea (Akkemik et al. 2005, 2008; Martin-Benitto et al. 2016), Mediterranean regions (Touchan et al. 2005a), as well as the Sivas (D'Arrigo & Cullen 2001), southwestern (Touchan et al. 2003, Touchan et al. 2007; Köse et al. 2013 ), south-central (Akkemik & Aras 2005) and western Anatolian (Köse et al. 2011) regions of Turkey. These studies used tree rings to reconstruct precipitation because available moisture is often found to be the most important limiting factor that influences radial growth of many tree species in Turkey. These studies revealed past spring-summer precipitation, and described past dry and wet events and their duration. Recently, Cook et al. (2015) presented Old World Drought Atlas (OWDA), which is a set of year by year maps of reconstructed Palmer Drought Severity Index from tree-ring chronologies over the Europe and Mediterranean Basin.

Besides detailed information on precipitation history represented by these paleoscientific studies, we have still very limited knowledge of past temperature variability of Turkey. For example, significant decreases in spring diurnal temperature ranges (DTR) occurred throughout Turkey from 1929 to 1999 (Turkes & Sumer 2004). This decrease in spring DTRs was characterized by day-time temperatures that remained relatively constant while a significant increase in night-time temperatures were recorded over western Turkey and were concentrated around urbanized and rapidly-urbanizing cities. The historical context of this gradual warming trend in spring temperatures is unclear. Heinrich et al. (2013) provided a winter-to-spring temperature proxy for Turkey from carbon isotopes within the growth rings of *Juniperus excelsa* M. Bieb. since AD

1125. Low-frequency temperature trends corresponding to the end of Medieval Climatic
Anomaly and Little Ice Age were identified in the record, but the proxy failed to identify the
recent warming trend during the 20[th] century. In this study, we present a tree-ring based spring
temperature reconstruction from Turkey and compare our results to previous reconstructions of
temperature and precipitation to provide a more comprehensive understanding of climate
conditions during the 19[th] and 20[th] centuries.

**2  Data and Methods**
2.1 Climate of the Study Area

The study area, which spans 36–42º N and 26–38º E, was based on the distribution of available
tree-ring chronologies. This vast area covers much of western Anatolia and includes the western
Black Sea, Marmara, and western Mediterranean regions. Much of this area is characterized by a
Mediterranean climate that is primarily controlled by polar and tropical air masses (Türkeş
1996a, Deniz et al. 2011). In winter, polar fronts from the Balkan Peninsula bring cold air that is
centered in the Mediterranean. Conversely, the dry, warm conditions in summer are dominated
by weak frontal systems and maritime effects. Moreover, the Azores high-pressure system in
summer and anticyclonic activity from the Siberian high-pressure system often cause below
normal precipitation and dry sub-humid conditions over the region (Türkeş 1999, Deniz et al.
2011). In this Mediterranean climate, annual mean temperature and precipitation range from 3.6
°C to 20.1 °C and from 295 to 2220 mm, respectively, both of which are strongly controlled by
elevation (Deniz et al. 2011).

2.2 Development of tree-ring chronologies

To investigate past temperature conditions, we used a network of 23 tree-ring site chronologies
(Fig. 1). Fifteen chronologies were produced by previous investigations (Mutlu et al. 2011,
Akkemik et al. 2008, Köse et al. unpublished data, Köse et al. 2011, Köse et al. 2005) that
focused on reconstructing precipitation in the study area. In addition, we sampled eight new
study sites and developed tree-ring time series for these areas (Table 1). Increment cores were
taken from living *Pinus nigra* Arnold and *Pinus sylvestris* L. trees and cross-sections were taken
from *Abies nordmanniana* (Steven) Spach and *Picea orientalis* (L.) Link trunks.

Samples were processed using standard dendrochronological techniques (Stokes & Smiley 1968,
Orvis & Grissino-Mayer 2002, Speer 2010).  Tree-ring widths were measured, then visually
crossdated using the list method (Yamaguchi 1991). We used the computer program COFECHA,
which uses segmented time-series correlation techniques, to statistically confirm our visual
crossdating (Holmes 1983, Grissino-Mayer 2001).  Crossdated tree-ring time series were then
standardized by fitting a 67% cubic smoothing spline with a 50% cutoff frequency to remove
non-climatic trends related to the age, size, and the effects of stand dynamics using the ARSTAN
program (Cook 1985, Cook et al. 1990a). These detrended series were then pre-whitened with
low-order autoregressive models to produce time series with a strong common signal and
without biological persistence. These series may be more suitable to understand the effect of
climate on tree-growth, even if any persistence due to climate might be removed by pre-
whitening. For each chronology, the individual series were averaged to a single chronology by
computing the biweight robust means to reduce the influences of outliers (Cook et al. 1990b). In
this research we used residual chronologies obtained from ARSTAN to reconstruct temperature.

The mean sensitivity, which is a metric representing the year-to-year variation in ring width
(Fritts 1976), was calculated for each chronology and compared. The minimum sample depth for
each chronology was determined according to expressed population signal (EPS), which we used
as a guide for assessing the likely loss of reconstruction accuracy. Although arbitrary, we
required the commonly considered threshold of EPS > 0.85 (Wigley et al. 1984; Briffa & Jones

140  1990).


2.3 Identifying relationship between tree-ring width and climate

We extracted high resolution monthly temperature and precipitation records from the climate
dataset CRU TS 3.23 gridded at 0.5º intervals (Jones and Harris 2008) from KNMI Climate
Explorer (http://climexp.knmi.nl) for 36–42 ºN, 26–38 ºE. The period AD 1930–2002 was
chosen for the analysis because it maximized the number of station records within the study area.

First, the climate-growth relationships were investigated with response function analysis (RFA)
(Fritts 1976) for biological year from previous October to current October using the
DENDROCLIM2002 program (Biondi & Waikul 2004). This analysis is done to determine the
months during which the tree-growth is the most responsive to temperature. RFA results showed
that precipitation from May to August and temperature in March and April have dominant
control on tree-ring formation in the area. Second, we produced correlation maps showing

155 correlation coefficients between tree-ring chronologies and the climate factors most important

156 for tree growth, which are May–August precipitation and March–April temperature, to find the

157 spatial structure of radial growth-climate relationship (St. George 2014, St. George and Ault

158 2014, Hellmann et al. 2016). For each site we used the closest gridded temperature and

159 precipitation values.

160

161   2.4 Temperature reconstruction

162

163 The climate reconstruction is performed by regression based on the principal component (PCs)

164 of the 23 chronologies within the study area. Principle Component Analysis (PCA) was done

165 over the entire period in common to the tree-ring chronologies.  The significant PCs were

166 selected by stepwise regression. We combined forward selection with backward elimination

167 setting $p < 0.05$ as entrance tolerance and $p < 0.10$ as exit tolerance. The final model obtained

168 when the regression reaches a local minimum of the root mean squared error (RMSE). The order

169 of entry of the PCs into the model was $PC_3$, $PC_{21}$, $PC_4$, $PC_{15}$, $PC_5$, $PC_{17}$, $PC_7$, $PC_9$, $PC_{10}$. The

170 regression equation is calibrated on the common period (1930–2002) between robust temperature

171 time-series and the selected tree-ring series. Third, the final reconstruction is based on bootstrap

172 regression (Till and Guiot 1990), a method designed to calculate appropriate confidence intervals

173 for reconstructed values and explained variance even in cases of short time-series. It consists in

174 randomly resampling the calibration datasets to produce 1000 calibration equations based on a

175 number of slightly different datasets.


The quality of the reconstruction is assessed by a number of standard statistics. The overall
quality of fit of reconstruction is evaluated based on the determination coefficient ($R^2$), which
expresses the percentage of variance explained by the model and RMSE, which expresses the
calibration error. This does not insure the quality of the extrapolation which needs additional
statistics based on independent observations, i.e. observations not used by the calibration
(verification data). They are provided by the observations not resampled by the bootstrap
process. The prediction RMSE (called RMSEP), the reduction of error (RE) and the coefficient
of efficiency (CE) are calculated on the verification data and enable to test the predictive quality
of the calibrated equations (Cook et al. 1994). Traditionally, a positive RE or CE values means a
statistically significant reconstruction model, but bootstrap has the advantage to produce
confidence intervals for such statistics without theoretical probability distribution and finally we
accept the RE and CE for which the lower confidence margin at 95% are positive. This is more
constraining than just accepting all positive RE and CE. For additional verification, we also
present traditional split-sample procedure results that divided the full period into two subsets of
equal length (Meko and Graybill 1995).

To identify the extreme March–April cold and warm events in the reconstruction, standard
deviation (SD) values were used.  Years one and two SD above and below the mean were
identified as warm, very warm, cold, and very cold years, respectively. As a way to assess the
spatial representation of our temperature reconstruction, we conducted a spatial field correlation
analysis between reconstructed values and the gridded CRU TS3.23 temperature field (Jones and
Harris 2008) for a broad region of the Mediterranean over the entire instrumental period (ca.
1930–2002).   Finally, we compared our temperature reconstruction and also precipitation signal
(PC1) against existing gridded temperature and hydroclimate reconstructions for Europe over the
period 1800–2002. We performed spatial correlation analysis between [1] our temperature
reconstruction and gridded temperature reconstructions for Europe (Xoplaki et al. 2005,
Luterbacher et al. 2016) and OWDA (Cook et al. 2015); and [2] PC1 and summer precipitation
reconstruction (Pauling et al., 2006) and Old World Drought Atlas (OWDA) (Cook et al. 2015).
To assess the significance of the correlation between our reconstruction (y) and gridded
reconstructions (xj, j=1…N), we have calculated significance thresholds based on a Monte-Carlo
technique. For each gridpoint j, we have calculated the correlation between xj and y, but with a
random permutation of the values of our reconstruction. This is repeated 1000 times with a
different permutation. The 1000 correlation coefficients so obtained are expected to be zero as
the correlation is established on non corresponding years. The 95th quantile of these 1000
coefficients is assumed to be passed in less than 5% of the cases. Then a correlation coefficient
with a higher value is considered as positive with a 95% confidence. These thresholds are
obtained with a common permutation for all xj so that the spatial structure is conserved in the
tests. The sign + is assigned to the xj with a correlation higher than an expected value under the
non-correlation hypothesis.

**2  Results and Discussion**
2.1 Tree-ring chronologies
In addition to 15 chronologies developed by previous studies, we produced six *P. nigra*, one *P.*
*sylvestris*, one *A. nordmanniana / P. orientalis* chronologies for this study (Table 2). The Çorum
district produced two *P. nigra* chronologies: one the longest (KAR; 627 years long) and the other
the most sensitive to climate (SAH; mean sensitivity value of 0.25). Previous investigations of
climate-tree growth relationships reported a mean sensitivity range of 0.13–0.25 for *P. nigra* in
Turkey (Köse 2011, Akkemik et al. 2008). The KAR, SAH, and ERC chronologies (with mean
sensitivity values from 0.22 to 0.25) were classified as very sensitive, and the SAV, HCR, and
PAY chronologies (mean sensitivity values range 0.17–0.18) contained values characteristic of
being sensitive to climate. The lowest mean sensitivity value was obtained for the ART *A.*
*nordmanniana / P. orientalis* chronology. Nonetheless, this chronology retained a statistically
significant temperature signal ($p < 0.05$).

2.2 Tree-ring growth-climate relationship
RFA coefficients of May to August precipitation are positively correlated with most of the tree-
ring series (Fig. 2) and among them, May and June coefficients are generally significant. The
first principal component of the 23 chronologies, which explains 47% of the tree-growth
variance, is highly correlated with May–August total precipitation, statistically ($r = 0.65$, $p <$
0.001) and visually (Fig. 3). The high correlation was expected given that numerous studies also
found similar results in Turkey (Akkemik 2000a, Akkemik 2000b, Akkemik 2003, Akkemik et
al. 2005, Akkemik et al. 2008, Akkemik & Aras 2005, Hughes et al. 2001, D'Arrigo &  Cullen
2001, Touchan et al. 2003; Touchan et al. 2005a, Touchan et al. 2005b, Touchan et al. 2007,
Köse et al. 2011, Köse et al. 2012,  Köse et al. 2013, Martin-Benitto et al. 2016). The influence
of temperature was not as strong as May–August precipitation on radial growth, although
generally positive in early spring (March and April) (Fig. 2). Conversely, the ART chronology
from northeastern Turkey contained a strong temperature signal, which was significantly positive
in March.
Correlation maps representing influence of May-August precipitation (Fig. 4a) and March-April
temperature (Fig 4b) also showed that strength of the summer precipitation signal is higher and
significant almost all over the Turkey. Higher precipitation in summer has a positive effect on
tree-growth, because of long-lasting dry and warm conditions over the Turkey (Türkeş 1996b,
Köse et al. 2012). Spring precipitation signal are generally positive and significant only for four
tree-ring sites. The sites located at the upper distributions of the species are generally showed
higher correlations. The highest correlations obtained for *Picea/Abies* chronology (ART) from
the Caucasus, and for *Pinus nigra* chronology (HCR) from the upper (about 1900 m) and
southeastern distribution of the species. This black pine forest was still partly covered by snow
from previous year during the field work in fall. Higher temperatures in spring maybe cause
snow melt earlier and lead to produce larger annual rings. In addition to these chronologies, we
also used the chronologies that revealed the influence of precipitation, as well as temperature to
reconstruct March–April temperature.


2.3 March-April temperature reconstruction
The higher order PCs of the 23 chronologies are significantly correlated with the March–April
temperature and, by nature, are independent on the precipitation signal (Table 3). The best
selection for fit temperature are obtained with the $PC_3$, $PC_4$, $PC_5$, $PC_7$, $PC_9$, $PC_{10}$, $PC_{15}$, $PC_{17}$,
$PC_{21}$, which explains together 25% of the tree-ring chronologies. So the temperature signal
remains important in the tree-ring chronologies and can be reconstructed. The advantage to
separate both signals through orthogonal PCs enable to remove an unwanted noise for our
temperature reconstruction. Thus, $PC_1$ was not used as potential predictor of temperature because
it is largely dominated by precipitation (Table 3, Fig. 3). The last two PCs contain a too small
part of the total variance to be used in the regressions. However, even if Jolliffe (1982) and Hadi
& Ling (1998) claimed that certain PCs with small eigenvalues (even the last one), which are
commonly ignored by principal components regression methodology, may be related to the
independent variable, we must be cautious with that because they may be much more dominated
by noise than the first ones. So, the contribution of each PC to the regression sum of squares is
also important for selection of PCs (Hadi & Ling 1998). The findings of Jolliffe (1982) and Hadi
& Ling (1998) provide a justification for using non-primary PCs, (*e.g*., of second and higher
order) in our regression, given that correlations with temperature may be over-powered by
affects from precipitation in our study area (Cook 2011, personal communication).

Using this method, the calibration and verification statistics indicated a statistically significant
reconstruction (Table 4, Fig. 5). For additional verification, we also present split-sample
procedure results. Similarly bootstrap results, the derived calibration and verification tests using
this method indicated a statistically significant RE and CE values (Table 5).

The regression model accounted for 67% (Adj. $R^2 = 0.64$, $p < 0.0001$) of the actual temperature
variance over the calibration period (1930–2002). Also, actual and reconstructed March–April
temperature values had nearly identical trends during the period 1930–2002 (Fig. 5). Moreover,
the tree-ring chronologies successfully simulated both high frequency and warming trends in the
temperature data during this period. The reconstruction was more powerful at classifying warm
events rather than cold events. Over the last 73 years, eight of ten warm events in the
instrumental data were also observed in the reconstruction, while five of nine cold events were
captured. Similarly, previous tree-ring based precipitation reconstructions for Turkey (Köse et al.
2011; Akkemik et al. 2008) were generally more successful in capturing dry years rather than
wet years.

Our temperature reconstruction on the 1800–2002 period is obtained by bootstrap regression,
using 1000 iterations (Fig. 6). The confidence intervals are obtained from the range between the
2.5$^{th}$ and the 97.5$^{th}$ percentiles of the 1000 simulations. Low frequency variability of our spring
temperature reconstruction showed larger variability in nineteenth century than twentieth
century. For the pre-instrumental period (1800–1929), a total of 23 cold (1813, 1818, 1821,
1824, 1837, 1848, 1854, 1858, 1860, 1869, 1877–1878, 1880–1881, 1883, 1897–1898, 1905–
1907, 1911–1912, 1923) and 13 warm (1801–1802, 1807, 1845, 1853, 1866, 1872–1873, 1879,
1885, 1890, 1901, 1926) events were determined. After comparing our results with event years
obtained from May–June precipitation reconstructions from western Anatolia (Köse et al. 2011),
the cold years 1818, 1848, and 1897 appeared to coincide with wet years and 1881 was a very
wet year for the entire region. Furthermore, these years can be described as cold (in March–
April) and wet (in May–June) for western Anatolia.

Among the warm periods in our reconstruction, conditions during the year 1879 were dry, 1895
wet, and 1901 very wet across the broad region of western Anatolia (Köse et al. 2011). Hence,
we defined 1879 as a warm (in March–April) and dry year (in May–June), and 1895 and 1901
were warm and wet years. In the years 1895 and 1901 the combination of a warm early spring
and a wet late spring-summer caused enhanced radial growth in Turkey, interpreted as longer
growing seasons without drought stress.

Of these event years, 1897 and 1898 were exceptionally cold and 1845, 1872 and 1873 were
exceptionally warm. During the last 200 years, our reconstruction suggests that the coldest year
was 1898 and the warmest year was 1873. The reconstructed extreme events also coincided with
accounts from historical records. Server (2008) recounted the winter of 1898 as characterized by
anomalously cold temperatures that persisted late into the spring season. A family, who brought
their livestock herds up into the plateau region in Kırşehir seeking food and water were suddenly
covered in snow on 11 March 1898. This account of a late spring freeze supports the
reconstruction record of spring temperatures across Turkey, and offers corroboration to the
quality of the reconstructed values.

Seyf (1985) reported that extreme summer temperature during the year 1873 resulted in
widespread crop failure and famine. Historical documents recorded an infamous drought-derived
famine that occurred in Anatolia from 1873 to 1874 (Quataert, 1996, Kuniholm, 1990), which
claimed the lives of 250,000 people and a large number of cattle and sheep (Faroqhi, 2009). This
drought caused widespread mortality of livestock and depopulation of rural areas through human
mortality, and migration of people from rural to urban areas. Further, the German traveler
Naumann (1893) reported a very dry and hot summer in Turkey during the year 1873 (Heinrich
et al, 2013). Conditions worsened when the international stock exchanges crashed in 1873
(Zürcher, 2004). Our temperature record suggests that dry conditions during the early 1870s
were possibly exacerbated by warm spring temperatures that likely carried into summer. A
similar pattern of intensified drought by warm temperatures was demonstrated recently by
Griffin and Anchukaitis (2014) for the current drought in California, USA.

Extreme cold and warm events were usually one year long, and the longest extreme cold and
warm events were two and three years, respectively.  These results were similar with durations of
extreme wet and dry events in Turkey (Touchan et al. 2003, Touchan et al. 2005a, Touchan et al.
2005b, Touchan et al. 2007, Akkemik & Aras, 2005, Akkemik et al. 2005, Akkemik et al. 2008,
Köse et al. 2011, Güner et al. 2016). Moreover, seemingly innocuous short-term warm events,
such as the 1807 event, were recorded across the Mediterranean and in high elevations of the
European regions. Casty et al. (2005) reported the year 1807 as being one of the warmest alpine
summers in the European Alps over the last 500 years. As such, a drought record from Nicault et
al. (2008) echoes this finding, as a broad region of the Mediterranean basin experienced drought
conditions.

~~Low frequency variability of our spring temperature reconstruction showed larger variability in~~
~~nineteenth century than twentieth century. Similar results observed on previous tree-ring based~~
~~precipitation reconstructions from Turkey (Touchan et al. 2003, D'Arrigo et al. 2001, Akkemik~~
~~and Aras 2005, Akkemik et al. 2005, Köse et al. 2011). Moreover, cold (warm) periods observed~~
~~in our reconstruction are generally appeared as generally wet in the precipitation reconstructions,~~
~~while rarely correlated with dry (wet) periods (Fig. 7).  When we compare the relationship~~
~~between temperature and precipitation over the instrumental period, both case, cold (warm) and~~
~~wet (dry) as well as cold (warm) and dry (wet), can be observed.~~

Heinrich et al. (2013) analyzed winter-to-spring (January–May) air temperature variability in
Turkey since AD 1125 as revealed from a robust tree-ring carbon isotope record from *Juniperus*
*excelsa*. Although they offered a long-term perspective of temperature over Turkey, the
reconstruction model, which covered the period 1949–2006, explained 27% of the variance in
temperature since the year 1949. In this study, we provided a short-term perspective of
temperature fluctuation based on a robust model (calibrated and verified 1930–2002; Adj. $R^2 =$
0.64; $p < 0.0001$). Yet, the Heinrich et al. (2013) temperature record did not capture the 20[th]
century warming trend as found elsewhere (Wahl et al. 2010). However, their temperature trend
does agree with trend analyses conducted on meteorological data from Turkey and other areas in
the eastern Mediterranean region. The warming trend seen during our reconstruction calibration
period (1930–2002) was similar to the data shown by Wahl et al. (2010) across the region and
hemisphere. Further, the warming trends seen in our record agrees with data presented by Turkes
& Sumer (2004), of which they attributed to increased urbanization in Turkey. Considering long-
term changes in spring temperatures, the 19[th] century was characterized by more high-frequency
fluctuations compared to the 20[th] century, which was defined by more gradual changes and
includes the beginning of decreased DTRs in the region (Turkes & Sumer 2004).

4 2.4 Comparison with instrumental gridded data and spatial reconstructions

Spatial correlation analysis revealed that our network-based temperature reconstruction was
representative of conditions across Turkey, as well as the broader Mediterranean region (Fig.
7 8). During the period 1930–2002, estimated temperature values were highly significant (*r* range
0.5–0.6, $p < 0.01$) with instrumental conditions recorded from southern Ukraine to the west
across Romania, and from northern areas of Libya and Egypt to the east across Iraq. The strength
of the reconstruction model is evident in the broad spatial implications demonstrated by the
temperature record. Thus, we interpret warm and cold periods and extreme events within the
record with high confidence.

We compared our tree-ring based temperature reconstruction with existing gridded temperature
reconstructions for Europe (Xoplaki et al. 2005, Luterbacher et al. 2016) and the Old World
Drought Atlas (OWDA) (Cook et al. 2015) for further validation of the reconstruction (Fig. 89a,
b, c, respectively). Spatial correlations over the past 200 years were lower with reconstructed
European summer temperature (May to July) (Fig. 89b). Yet, we expected this result because of
the paucity of Turkey-derived proxies in the other reconstructions, as well as the differing
seasons involved across the reconstructions. Similarly, our reconstruction showed weak
correlations with summer drought index over Turkey. Beside comparing different seasons,
perhaps this is because less precipitation begets drought conditions rather than high temperature
in the region. The highest and significant ($p < 0.05$) correlations were found with European
spring (March to May) temperature reconstruction over southeastern Europe, which are stronger
over Turkey (Fig. 89a). We used the mean of corresponding grid points from European spring
temperature reconstruction over the study area (36–42º N, 26–38º E) to show how the correlation
changed over time (Fig. 910).  The correlation coefficient was highly significant ($r = 0.76$, $p <$
0.001) during our calibration period (1930–2002). We found lower but still significant
correlation ($r = 0.35$, $p < 0.10$) for the period of 1901–1929, which climatic records are very few
over the region while available data has sufficient quality for most part of Europe. These results
give additional verification for our reconstruction. Moreover, our reconstruction has a weak,
insignificant relationship ($r = 0.13$, $p > 0.10$) during the 19[th] century. This may be related to poor
reconstructive skill of European spring temperature reconstruction over Turkey, which contains
few proxies from the country (Xoplaki et al. 2005, Luterbacher et al. 2004). Nonetheless, these
results demonstrate that tree-ring chronologies from Turkey can serve as useful temperature
proxies for further spatial temperature reconstructions to fill the gaps in the area.

We also compared the precipitation signal (PC1) obtained from our tree-ring network with Old
World Drought Atlas (OWDA) (Cook et al. 2015) and gridded European summer precipitation
reconstruction (Pauling et al., 2006) to test the strength of the signal spatially (Fig. 89d and e,
respectively). We calculated highly significant positive correlations with summer drought index
over Turkey and neighboring European countries such as Greece, Bulgaria, and Romania, Italy
while significant correlations are lowver for the other northern Mediterranean countries (Fig.
89d). These results showed that summer precipitation signal represented by PC1 is very strong
not only on instrumental period, but also on pre-instrumental period, and represents a large
spatial coverage. We found lowver but stilland in-significant correlations over Turkey and
Mediterranean countries with European summer precipitation reconstruction (Fig. 89e). Pauling
et al. (2006) stated that poor reconstructive skills determined over Turkey because of few
instrumental record before the1930s. These results showed that summer precipitation signal
represented by PC1 is very strong not only on instrumental period, but also on pre-instrumental
period, and represents a large spatial coverage.

**4 Conclusions**

In this study, we used a broad network of tree-ring chronologies to provide the first tree-ring
based temperature reconstruction for Turkey and identified extreme cold and warm events during
the period 1800–1929 CE. Similar to the precipitation reconstructions against which we compare
our air temperature record, extreme cold and warm years were generally short in duration (one
year) and rarely exceeded two-three years in duration. The coldest and warmest years over
western Anatolia were experienced during the 19th century, and the 20th century is marked by a
temperature increase.

Reconstructed temperatures for the 19th century suggest that more short-term fluctuations
occurred compared to the 20th century. The gradual warming trend shown by our reconstruction
calibration period (1930–2002) is coeval with decreases in spring DTRs. Given the results of
Turkes and Sumer (2004), the variations in short- and long-term temperature changes between
the 19th and 20th centuries might be related to increased urbanization in Turkey.

We highlight that the 20th century warming trend is unprecedented within the context of the past
*ca.* 200 years, especially over the past *ca.* 15 years. Correlations with gridded climate fields and
other climate reconstructions from the region revealed that our network-based temperature
reconstruction was representative of conditions across Turkey, as well as the broader
Mediterranean region. Expanding the tree-ring network across Turkey, especially to the east, will
improve the spatial implications of future temperature reconstructions.

The study revealed the potential for reconstructing temperature in an area previously thought
impossible, especially given the strong precipitation signals displayed by most tree species
growing in the dry Mediterranean climate that characterizes broad areas of Turkey. Our
reconstruction only spans 205 years due to the shortness of the common interval for the
chronologies used in this study, but the possibility exists to extend our temperature
reconstruction further back in time by increasing the sample depth with more temperature-
sensitive trees, especially from northeastern Turkey. Thus future research will focus on
increasing the number of tree-ring sites across Turkey, and maximizing chronology length at
existing sites that would ultimately extend the reconstruction back in time.

**Acknowledgements**

This research was supported by The Scientific and Technical Research Council of Turkey
(TUBITAK); Projects ÇAYDAG 107Y267 and YDABAG 102Y063. N. Köse was supported by
The Council of Higher Education of Turkey. We are grateful to the Turkish Forest Service
personnel and Ali Kaya, Umut Ç. Kahraman and Hüseyin Yurtseven for their invaluable support
during our field studies. We thank to Dr. Ufuk Turuncoğlu for his help on spatial analysis. J.
Guiot was supported by the Labex OT-Med (ANR-11-LABEX-0061), French National Research
Agency (ANR).

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

Table 1. Site information for the new chronologies developed by this study in Turkey.

| Site name | Site code | Species | No. trees/ cores | Aspect | Elev. (m) | Lat. (N) | Long. (E) |
|---|---|---|---|---|---|---|---|
| Çorum, Kargı, Karakise kayalıkları | KAR | *Pinus nigra* | 22 / 38 | SW | 1522 | 41°11' | 34°28' |
| Çorum, Kargı, Şahinkayası mevkii | SAH | *P. nigra* | 12 / 21 | S | 1300 | 41°13' | 34°47' |
| Bilecik, Muratdere | ERC | *P. nigra* | 12 / 25 | SE | 1240 | 39°53' | 29°50' |
| Bolu, Yedigöller, Ayıkaya mevkii | BOL | *P. sylvestris* | 10 / 20 | SW | 1702 | 40°53' | 31°40' |
| Eskişehir, Mihalıççık, Savaş alanı mevkii | SAV | *P. nigra* | 10 / 18 | S | 1558 | 39°57' | 31°12' |
| Kayseri, Aladağlar milli parkı, Hacer ormanı | HCR | *P. nigra* | 18 / 33 | S | 1884 | 37°49' | 35°17' |
| Kahramanmaraş, Göksun, Payanburnu mevkii | PAY | *P. nigra* | 10 / 17 | S | 1367 | 37°52' | 36°21' |
| Artvin, Borçka, Balcı işletmesi | ART | *Abies nordmanniana Picea orientalis* | 23 / 45 | N | 1200– 2100 | 41°18' | 41°54' |


Table 2. Summary statistics for the new chronologies developed by this study in Turkey.

| | | Total chronology | | | Common interval | |
|---|---|---|---|---|---|---|
| Site Code | Time span | 1st year (*EPS > 0.85) | Mean sensitivity | Time span | Mean correlations: among radii /between radii and mean | Variance explained by PC1 (%) |
| KAR | 1307–2003 | 1620 | 0.22 | 1740–1994 | 0.38 / 0.63 | 41 |
| SAH | 1663–2003 | 1738 | 0.25 | 1799–2000 | 0.42 / 0.67 | 45 |
| ERC | 1721–2008 | 1721 | 0.23 | 1837–2008 | 0.45 / 0.69 | 48 |
| BOL | 1752–2009 | 1801 | 0.18 | 1839–1994 | 0.32 / 0.60 | 36 |
| SAV | 1630–2005 | 1700 | 0.17 | 1775–2000 | 0.33 / 0.60 | 38 |
| HCR | 1532–2010 | 1704 | 0.18 | 1730–2010 | 0.38 / 0.63 | 40 |
| PAY | 1537–2010 | 1790 | 0.18 | 1880–2010 | 0.28 / 0.56 | 32 |
| ART | 1498–2007 | 1624 | 0.12 | 1739–1996 | 0.37 / 0.60 | 41 |

*EPS = Expressed Population Signal [Wigley et al., 1984]

Table 3. Principal components analysis statistics for the Turkey temperature reconstruction
model.

| | Explained variance | Correlation coefficients with | | The chronologies represented by higher magnitudes** in the eigenvectors |
|---|---|---|---|---|
| | | May–August PPT | March–April TMP | |
| | (%) | | | |
| PC1 | 46.57 | 0.65 | 0.19 | KAR, KIZ, TEF, BON,USA,TUR, CAT, INC, ERC, YAU, SAV, TAN, SIU |
| PC2 | 7.86 | –0.07 | 0.15 | KAR, SAV, TIR, BOL, YAU, ESK, TEF,BON, SIU |
| PC3* | 4.93 | 0.04 | –0.48 | HCR, PAY, BOL, YAU, SIA |
| PC4* | 4.68 | 0.11 | 0.17 | TEF, KEL, FIR, SIA, KIZ, SIU, ART |
| PC5* | 4.42 | –0.25 | 0.27 | SAH, TIR, FIR, ART |
| PC6 | 3.73 | 0.15 | –0.14 | KIZ, FIR, SAV, KAR, TIR, PAY, ESK, TEF, BON, ART |
| PC7* | 3.56 | 0.19 | 0.18 | KIZ, BON, BOL, YAU, HCR, PAY, INC |
| PC8 | 2.87 | 0.26 | 0.01 | HCR, ESK, BON, FIR, ERC, SIA |
| PC9* | 2.45 | 0.16 | 0.17 | PAY, USA, BOL, YAU, TIR, HCR, FIR, SIA, SIU |
| PC10* | 2.21 | 0.14 | –0.08 | TUR, CAT, SAV, SIA, KEL, ERC, SIU |
| PC11 | 2.09 | –0.36 | –0.20 | HCR, TEF, USA, INC, PAY, TUR, SAV, SIU |
| PC12 | 1.80 | –0.12 | 0.05 | TEF, CAT, YAU HCR, ESK, USA, BOL, SIA |
| PC13 | 1.63 | –0.06 | 0.17 | TEF, TUR, BOL, KAR, YAU, SIA |
| PC14 | 1.55 | –0.14 | 0.06 | TIR, USA, FIR, TUR, YAU, KAR, BON |
| PC15* | 1.50 | –0.20 | –0.14 | KIZ, BON, USA, ESK, INC, BOL |
| PC16 | 1.31 | 0.04 | 0.08 | SAH, HCR, INC, YAU, SAV, KAR, FIR, BOL, SIU |
| PC17* | 1.25 | 0.15 | 0.19 | SAH, SIU, KAR, ESK, TUR, ERC |
| PC18 | 1.14 | 0.13 | 0.02 | KAR, TEF, TUR, SAV, BON, CAT |
| PC19 | 1.09 | 0.16 | –0.11 | PAY, INC, SAV, HCR, KEL, CAT, TAN |
| PC20 | 0.95 | –0.15 | –0.01 | TIR, SAH, CAT |
| PC21* | 0.89 | 0.06 | –0.28 | TUR, INC, TIR, SAV |
| PC22 | 0.85 | 0.44 | 0.10 | KIZ, SAH, BON, YAU, SIU |
| PC23 | 0.67 | –0.22 | –0.02 | TAN, KEL, TUR, CAT |

"*" indicates the PCs, which used in the reconstruction as predictors
"**" which exceed ±0.2 value.




Table 4. Calibration and verification statistics of bootstrap method (1000 iterations

applied) showing the mean values based on the 95% confidence interval (CI).

|  |  | Mean (95% CI) |
| --- | --- | --- |
| Calibration | RMSE | 0.65 (0.52; 0.77) |
|  | $R^2$ | 0.73 (0.60; 0.83) |
| Verification | RE | 0.54 (0.15; 0.74) |
|  | CE | 0.51 (0.04; 0.72) |
|  | RMSEP | 0.88 (0.67; 1.09) |

*RMSE* root mean squared error; $R^2$ coefficient of determination; *RE* reduction of error; *CE*

coefficient of efficiency; *RMSEP* root mean squared error prediction

Table 5. Calibration and cross-validation statistics for the Turkey temperature reconstruction
model.

| Calibration Period | Verification Period | Adj. $R^2$ | F | RE | CE |
| --- | --- | --- | --- | --- | --- |
| 1930–1966 | 1967–2002 | 0.55 | 5.91 | 0.64 | 0.58 |
|  |  |  | $p < 0.0001$ |  |  |
| 1967–2002 | 1930–1966 | 0.71 | 10.45 | 0.63 | 0.46 |
|  |  |  | $p < 0.0001$ |  |  |




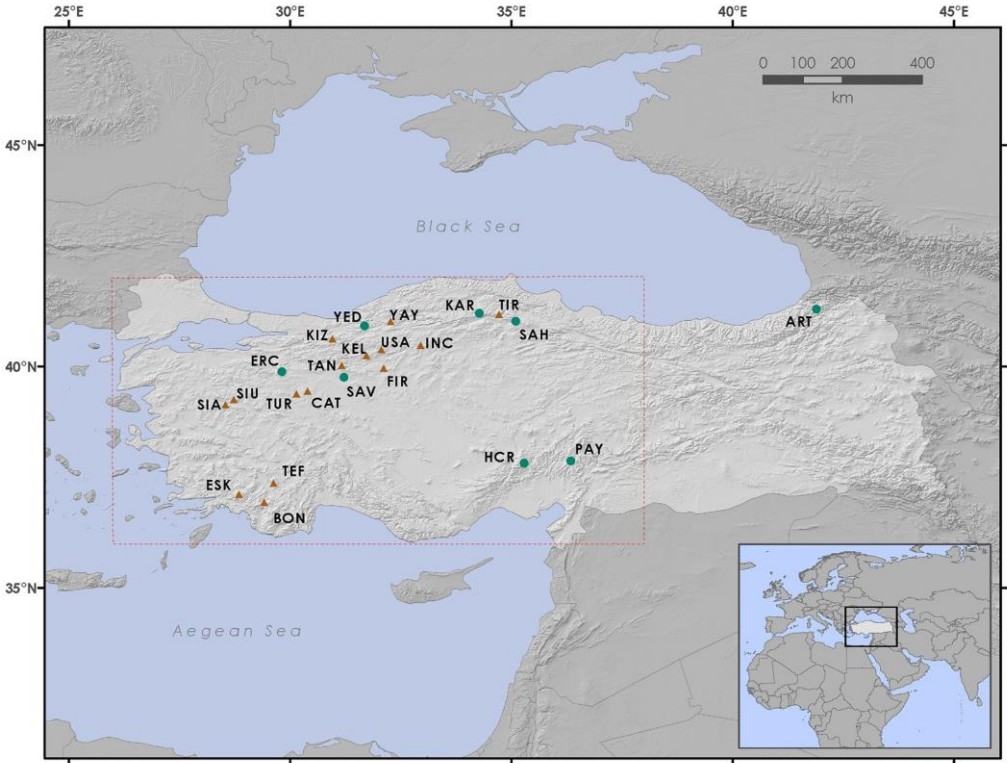

**Figure 1.** Tree-ring chronology sites in Turkey used to reconstruct temperature. Circles
represent the new sampling efforts from this study and the triangles represent previously-
published chronologies (YAY, SIA, SIU: Mutlu et al. 2011; TIR: Akkemik et al. 2008; TAN:
Köse et al. unpublished data; KIZ, ESK, TEF, BON, KEL, USA, FIR, TUR: Köse et al. 2011;
CAT, INC: Köse et al. 2005). The box (dashed line) represents the area for which the
temperature reconstruction was performed.



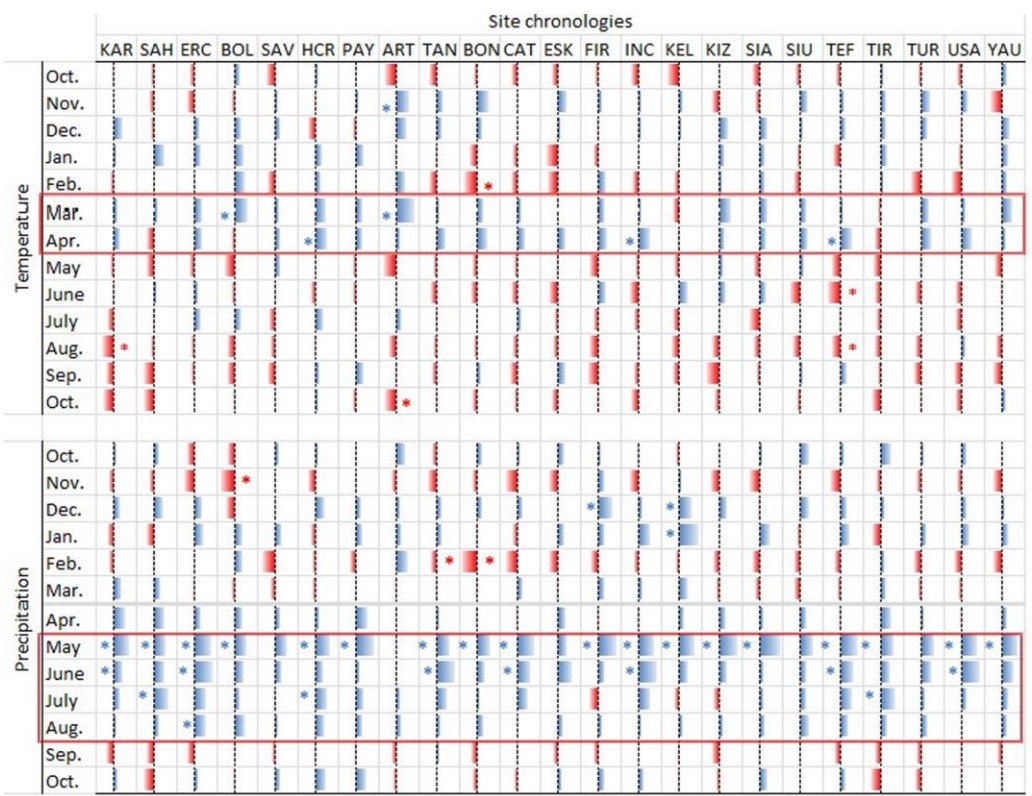


**Figure 2.** Summary of response function results of 23 chronologies. Red color represents

negative effects of climate variability on tree ring width; blue color represents positive effects of

climate variability on tree ring width. "*" indicates statistically significant response function

confidents ($p < 0.05$). Each response function includes 13 weights for average monthly

temperatures and 13 monthly precipitations from October of the prior year to October of current

year.

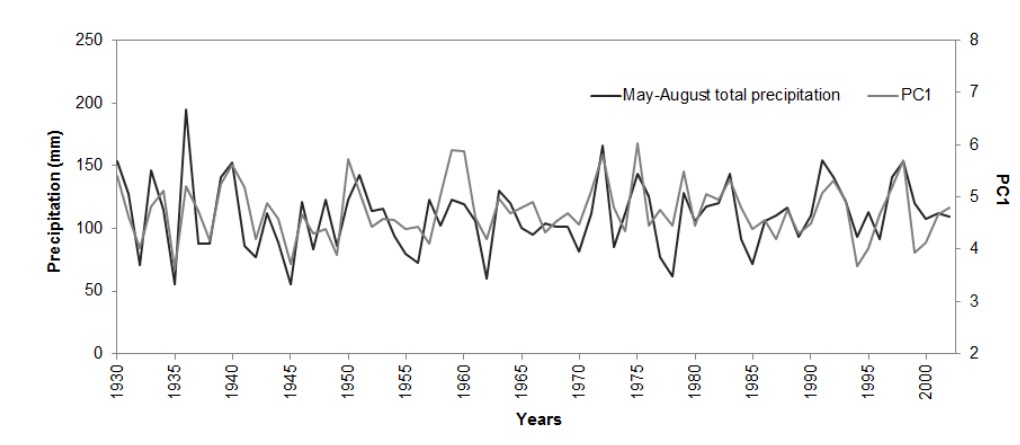


**Figure 3.** The comparison of May–August total precipitation (black) and the first principal
component of 23 tree-ring chronologies (gray). Correlation coefficient between two time series is
0.65 ($p < 0.001$).



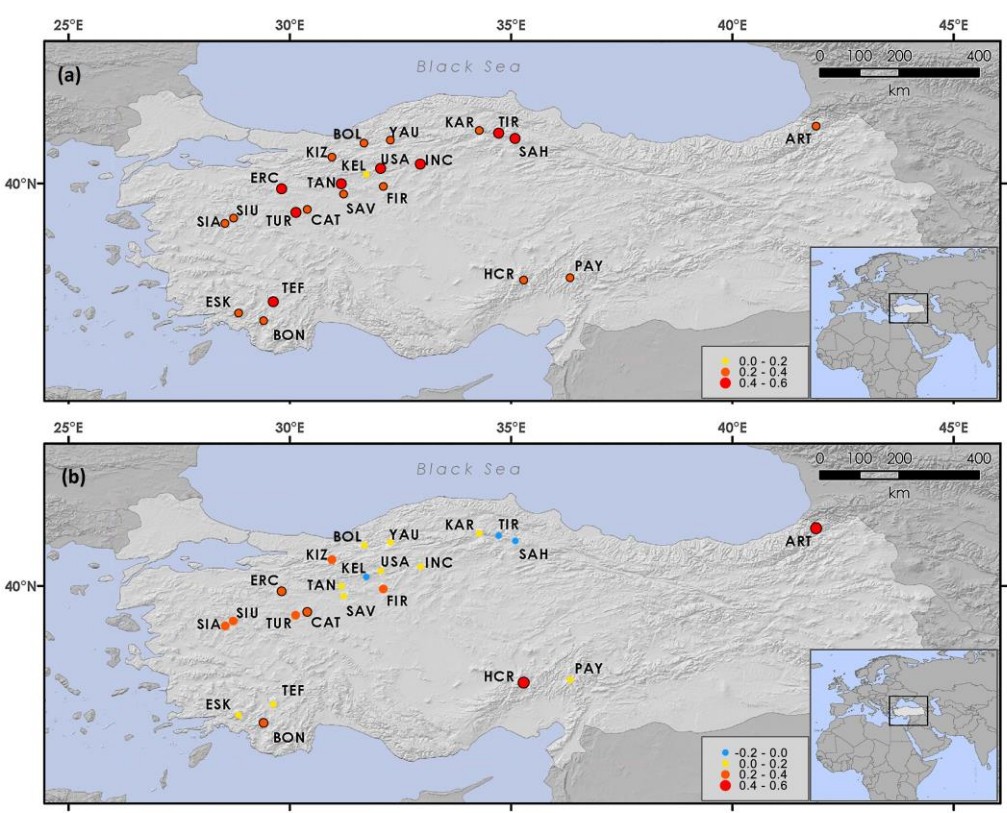


**Figure 4.** Maps showing Pearson's correlation coefficients between the sites chronologies and (a) May–August total precipitation and (b) March-April mean temperature for the period 1930–2012. For each site, the closest gridded (0.5° x 0.5°) climate data obtained from CRU dataset were used. Graduated circle size and color correspond to correlation coefficient versus the climate variable. Black lines surrounding circles represent significant correlation coefficients ($p < 0.05$).





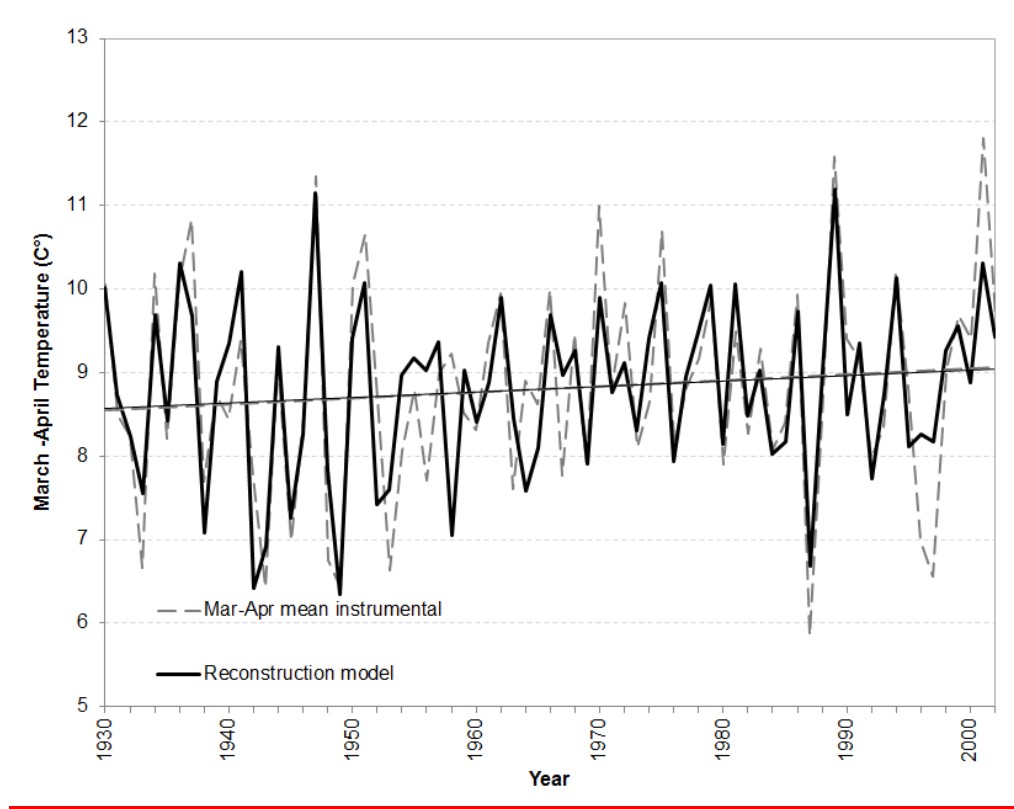


**Figure 5.** Actual (instrumental) and reconstructed March–April temperature (°C). Dashed lines

(dark grey) represent actual values and solid lines (black) represent reconstructed values shown

with trend lines (linear dashed grey and linear black lines, respectively). ~~Note: y axes labels~~

~~range 5–13 °C.~~The tendency to warm up at the reconstructed temperature is in good agreement

with the trend in instrumental data.









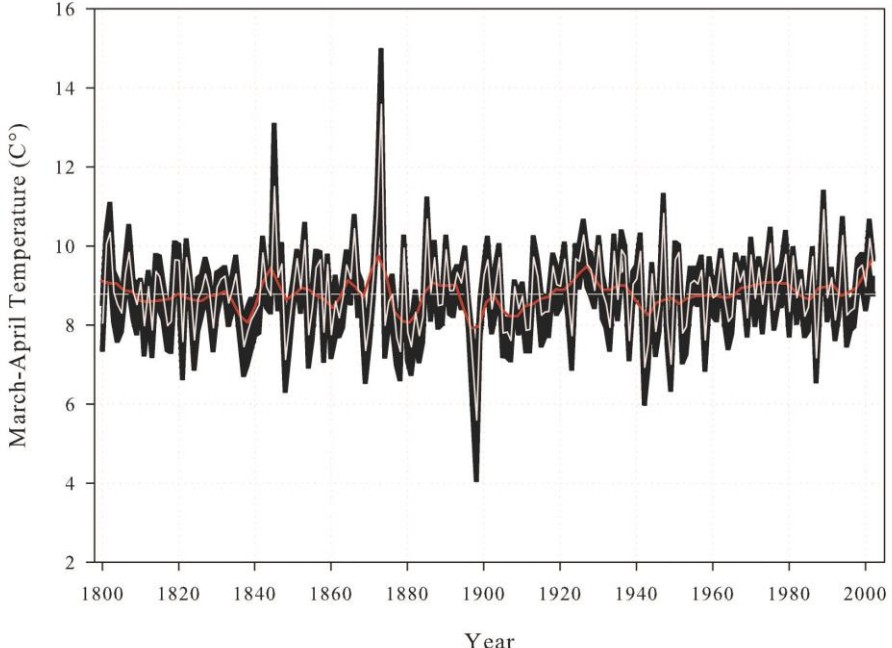


**Figure 6.** March–April temperature reconstruction for Turkey for the period 1800–2002

CE. The central horizontal line (dashed white) shows the reconstructed long-term mean and

does not include instrumental data; black background denotes Monte Carlo ($n = 1000$)

bootstrapped 95% confidence limits; and the red line shows 13-year low-pass filter values.

Note: y-axis labels range 2–16 °C.





**Figure 7.** Low frequency variability of previous tree-ring based precipitation
reconstructions from Turkey and spring temperature reconstruction. Each line shows 13-
year low-pass filter values. z scores were used for comparison.

Biçimlendirilmiş: Ortadan

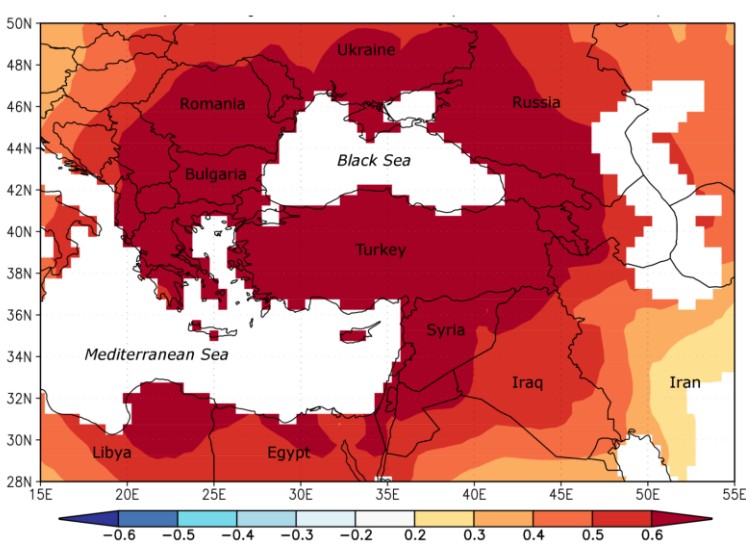


**Figure 78.** Spatial correlation map for the March–April temperature reconstruction. Spatial
field correlation map showing statistical relationship between the temperature
reconstruction and the gridded temperature field at 0.5º intervals (CRU TS3.23; Jones and
Harris 2008) during the period 1930–2002 over the Mediterranean region. For each grid,
calculated correlation coefficient from 0.20 to 0.60 is significant ($p < 0.05$).

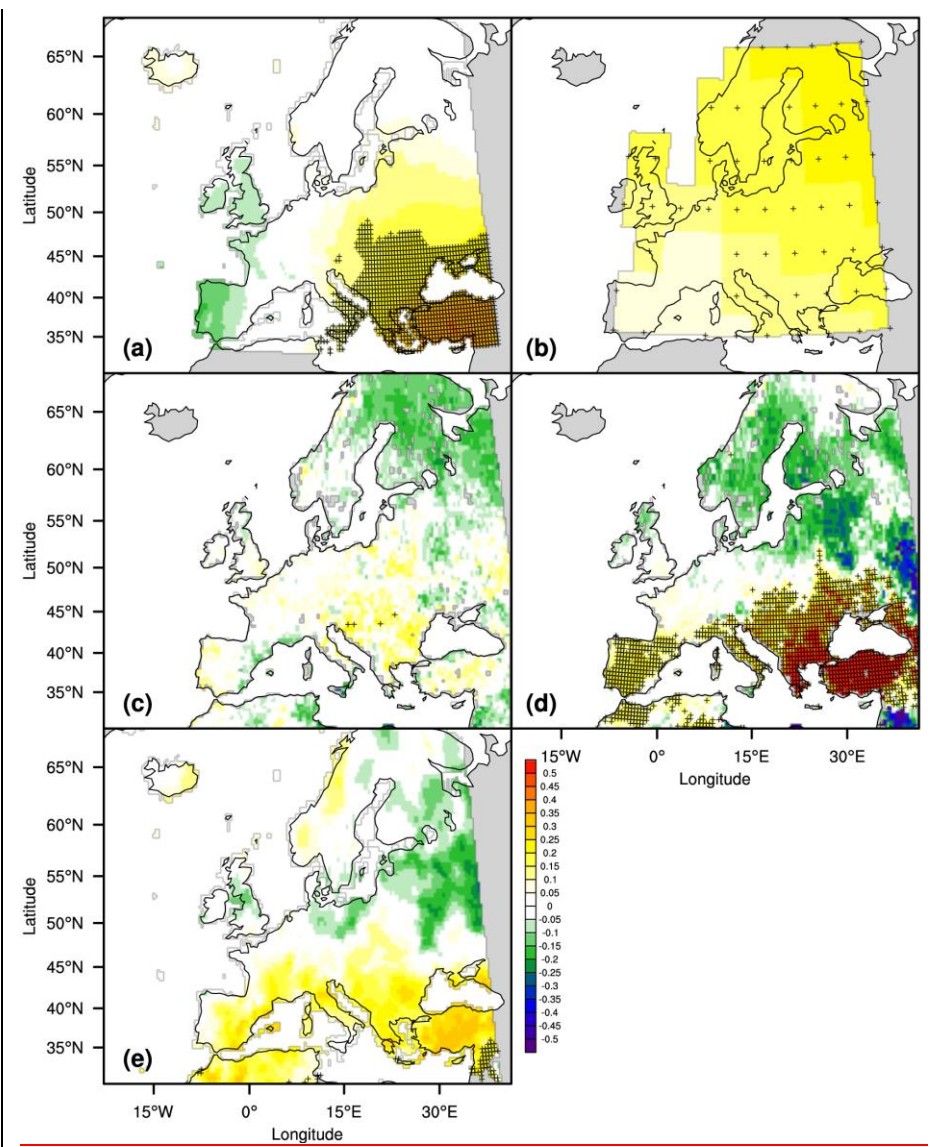



**Figure 89.** Spatial correlation maps for the March–April temperature reconstruction and
precipitation signal (PC1) obtained from tree-ring data set during the period 1800–2002 over
Europe. Maps demonstrate spatial field correlations between our temperature reconstruction and
(a) gridded spring temperature reconstruction for Europe (Xoplaki et al. 2005), (b) gridded
summer temperature reconstruction for Europe (Luterbacher et al. 2016), (c) Old World Drought
Atlas (OWDA; Cook et al. 2015). Panels (d) and (e) show spatial correlations between PC1 and
OWDA (Cook et al. 2015) and gridded European summer precipitation reconstruction (Pauling
et al., 2006), respectively. '+' represents significant correlation coefficients ($p < 0.05$).

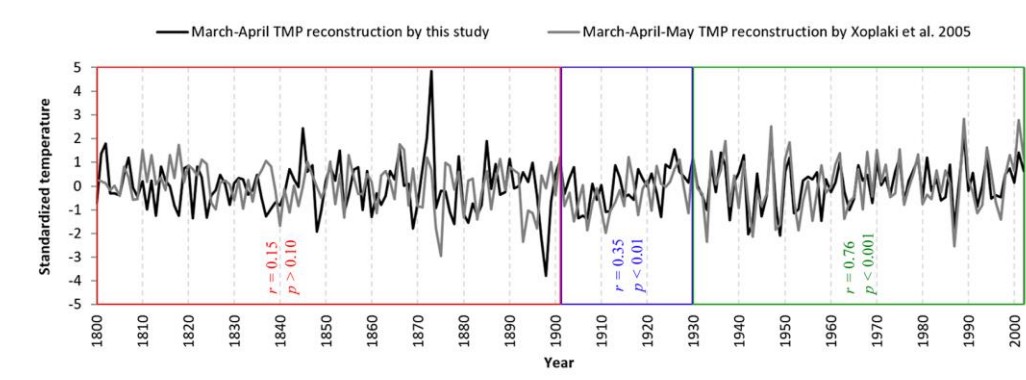




**Figure ~~10~~9.** Comparison of March-April temperature reconstruction (gray) with the mean
of corresponding grid points from European spring (March to May) temperature
reconstruction (Xoplaki et al. 2005; black) over the study area (36–42º N, 26–38º E). The
indicated correlation coefficients are calculated for instrumental period (also calibration
period for this study) (1930–2002; $r = 0.76$, $p < 0.001$); for the pre-instrumental period of
Turkey, while instrumental data has sufficient quality for most part of Europe (1901–1929;
$r = 0.35$, $p < 0.10$); and for pre-instrumental period (1800–1900; $r = 0.13$, $p < 0.10$).