# Peer review of "Spring temperature variability over Turkey since 1800 CE reconstructed"

_Climate of the Past, 2015_

## Referee Comment (RC1) · Anonymous Referee #1 · 29 Apr 2016

The paper presents a temperature reconstruction from Turkey over the past 200 years using 23 tree-ring chronologies.

General Comments:

I reviewed this paper before. I believe the authors submitted the same paper to the Italian Society of Silviculture and Forest Ecology in 2012. It is a modified version of the previous one. This is potentially an important research in reconstructing March-April temperature reconstruction for Turkey. The authors did an excellent job on developing the chronologies. However, I don't agree with the authors on several points in the manuscript.

[Figure]

One important question is why the authors did not develop their temperature reconstruction using only the chronologies that have significant relationship to temperature? A sensible approach would be to first screen the chronologies to remove those not significantly correlated with temperature. It seems to me they highly manipulated the data and used a very complicated equation to get a high adjusted R2.

Specific Comments:

1. Page 3, line 52-53. Hughes et al., (2011) did not develop any reconstruction, but they investigated the climate signal.

2. Page 5-lines 90-91: The authors should cite the investigators produced the chronologies.

3. Page 6-lines 128-129: What the authors mean by "Third, the final reconstruction is based 128 on bootstrap regression (Till and Guiot, 1990), the best method to assess the quality of the. . ." It is an awkward and not a scientific statement.

4. Page 7-line 143: another awkward sentence "but bootstrap is much more interesting. . ."

5. Pages 9-10, Temperature reconstruction: The authors mentioned that they conducted PCA on the 23 chronologies. I have several questions and comments on this section. The authors used stepwise regression (SR), however, they did not give enough details about this procedure. I am concerned that the model could be over-fitted and some of the predictors could be just noise. What criteria were applied to end the stepwise process (e.g., p-to-enter, p-to remove)? Was a conventional statistic such as Mallows' Cp used to arrive at the final model? Does the validation CE and RE continue to increase through each step of the stepwise? Did the authors run SR on each calibration period independently or use the same variables that were suggested by the SR for the whole period?

6. Page 7, section March-April temperature reconstruction. How did the chronologies

cluster around each pc that they used in their equation?

7. Conclusion, line 325-327: it is an awkward statement. Did any of previous authors indicated in any of their publications it is IMPOSSIBLE to reconstruct the temperature in the eastern Mediterranean. Did the authors read the mind of these authors?

8. Show the minor ticks in Figure 4, 5, and 7.

I don't recommend the paper at this stage.

---

## Referee Comment (RC2) · Anonymous Referee #2 · 7 May 2016

General comments: This is a nice paper overall and is worthy of publication because, as indicated in the Introduction, there are no long temperature records in Turkey for the assessment of changing springtime temperature warming, especially that related to a decrease in diurnal temperature range (DTR) through an increase in nighttime temperatures. This paper does not directly reconstruct springtime DTR, but it does produce the first springtime (March-April) temperature reconstruction for a region where it has not been done before. This is a useful result in and of itself. For this reason, I recommend publication after a few recommended changes and clarifications below are addressed.

Specific comments:

[Figure]

1) Discussion of how the tree-ring chronologies were produced for reconstruction (pg. 5) is adequate overall, but it sounds like the residual chronologies from ARSTAN were used for reconstruction. Is that true? If so isn't there a concern that a certain amount of low-frequency variability due to climate will be lost after prewhitening? This is a legitimate concern because it is not strictly true to say that "persistence not related to climate variations" (line 104) is only being removed from the ARSTAN residual chronology. If there is any persistence due to climate in the chronologies, it will be convolved with biological persistence ("persistence not related to climate") and therefore removed by prewhitening too. This is an important distinction that must be considered when and how to use autoregressive prewhitening. How this issue might affect the reconstruction presented here is unclear and should be at least mentioned.

2) Given that the 23 tree-ring chronologies used for temperature reconstruction are in general more precipitation sensitive (mainly May-June) than temperature sensitive (mainly March-April) (Fig. 2), there are reasonable grounds for concern that the temperature-only reconstruction will in fact be mixed with precipitation effects on tree growth. To reduce the likelihood of this being a problem, a stepwise PC regression procedure was used in which all 23 tree-ring PCs temperature were tested for use as predictors of Mar-Apr temperature, resulting in 9 PCs being entered into the model (Table 3). This appears to factor out and use the reasonably discrete March-April temperature signals in the chronologies quite well while minimizing the influence of precipitation in the final reconstruction. I would like a bit more information about this model, e.g. the order of entry of the PCs into the model and the stopping criterion for entering variables in the model. I also assume that the PCA applied to the chronologies was done over the entire period in common to the tree-ring chronologies, thus making the PCs non-orthogonal by some amount over the 1930-2002 calibration period and thus necessitating the use of stepwise regression. It would be good to clarify exactly what was done here.

3) Regarding the quality (skill) of the reconstruction, the resulting bootstrapped reconstruction and RE and CE statistics appear to support the reconstruction method used (Table 4). However, I don't believe this is the most powerful use of RE and CE because those statistics use the calibration and verification means, respectively, as the basis for assessing reconstruction skill typically against a block of withheld instrumental data. This implies that split calibration/verification may provide a more rigorous test then the way it is done with the bootstrap. Given this suggestion, it would be nice to see how good the RE and CE statistics are over the 1901-29 period of withheld temperature data as well in comparison to those provided in Table 4. The correlations look very good over this period (Fig. 6), so why not calculate the RE and CE too? Since RE and CE are more sensitive to changes in mean level between the calibration and verification periods, it could indicate how strong an effect there might be in using the assumed prewhitened chronologies as predictors. I would also like to see actual and reconstructed temperatures plotted in Fig. 4 extended back to 1901. All this with the understood caveat that the 1901-29 temperature data over Turkey are interpolated from surrounding areas.

4) Eq. (1) on page 10 is fine for showing the regression weightings of the selected PCs in the temperature reconstruction model, but this does not provide any useful information on the relative importance of the 23 tree-ring chronologies used in the reconstruction. This can, in the case of a fully orthogonal PC regression, be easily provided through the algebraic back-transformation of the regression model coefficients from PC space to tree-ring data space. It is unfortunate that this cannot be done here because (as best as I understand the method in the paper) the PCs in the calibration period will not be orthogonal by some unknown amount. Consequently, I am not sure the Eq. (1) is all that informative. It can be easily left out in my opinion.

A suggested reference to add: A recent paper on rainfall reconstruction that includes far eastern Turkey might be cited to: Martin-Benito, D., Ummenhofer C.C., Köse, N., Güner, H.T., Pederson, N. 2016. Tree-ring reconstructed May-June precipitation in the Caucasus since 1752 CE. Climate Dynamics

---

## Author Comment (AC1) · 17 May 2016

Thank you for your time and comments. We would like to thank you for your time and comments. Here we will comment on, one-by one, the referee comments/suggestions. Below each comment is our response in regular weight blue font.

Sincerely, Nesibe Köse

General Comments:

We could not use only the chronologies that have significant relationship to temperature, because at the same time they have significant precipitation signal (except ART chronology, Figure 2). On the other hand, we would like to show that it is possible to

make a climate reconstruction from a tree-ring network, even if this climate variable is not the most important limiting factor on radial growth. In our case, May to August precipitation was the most important factor, and the second one was March-April TMP for almost all the chronologies. Classical approach in Dendroclimatology, is to use the PC 1 and/or high order PCs reconstruct precipitation. But here, we would like show that PC 1 could be a signal for precipitation but a noise for temperature. On the other hand the other PC's, which explain less variance, could be noise for precipitation and but a signal for temperature.

Specific Comments: 1. Thank you for your attention we will correct it in the manuscript.

2. We cited the investigators produced the chronologies.

3. We will replace the sentence by: "Third, the final reconstruction is based on bootstrap regression (Till and Guiot, 1990), a method designed to calculate appropriate confidence intervals for reconstructed values and explained variance even in cases of short time-series."

4. We will replace by "... but bootstrap has the advantage to produce confidence intervals for such statistics without theoretical probability distribution and finally we accept the RE and CE for which the lower confidence margin at 95% are positive. This is more constraining than just accepting all positive RE and CE."

5. We added information in the text under the titles "Data and Method", "Temperature reconstruction" explaining which method we used stepwise regression. We combined forward selection with backward elimination, checking for entry, then removal, until no more variables can be added or removed. Each procedure requires only that we set significance levels (or critical values) for entry and/or removal. We used $p \leq 0.05$ as entrance tolerance and $p \leq 0.1$ as exit tolerance. Actually, for almost all PCs it was $p \leq 0.01$ in entire regression. The final model obtained when the regression reaches a local minimum of RMSE. We also calculated Mallows Cp values. See the relation Cp and p (the number of parameters in the model, including the intercept) in (Fig.1).

We did not used a split-sample procedure to verify the model stability. We used bootstrap method. Therefore we run SR for the whole period. Bootstrap is only applied to the selected set of predictors by stepwise regression. Then it is not concerned by the bootstrap. We did not calculated calulated RE, CE at each step of the stepwise regression.

6. We added a column to Table 3, to show the chronologies represented by higher magnitudes of the eigenvectors.

7. We tried to say with this sentence that no temperature reconstruction has been made, which mean that it is difficult to do that.

8. We did suggested changes in the figures.

Please also note the supplement to this comment:
http://www.clim-past-discuss.net/cp-2015-195/cp-2015-195-AC1-supplement.pdf

———————————————————

[Figure]

**Fig. 1.**

Figure showing a scatter plot of Cp versus p (the number of parameters in the model).

**Supplement:**

Thank you for your time and comments. We would like to thank you for your time and comments. Here we will comment on, one-by one, the referee comments/suggestions. Below each comment is our response in regular weight blue font.

Sincerely,
Nesibe Köse

One important question is why the authors did not develop their temperature reconstruction using only the chronologies that have significant relationship to temperature? A sensible approach would be to first screen the chronologies to remove those not significantly correlated with temperature. It seems to me they highly manipulated the data
and used a very complicated equation to get a high adjusted R2.

We could not use only the chronologies that have significant relationship to temperature, because at the same time they have significant precipitation signal (except ART chronology, Figure 2). On the other hand, we would like to show that it is possible to make a climate reconstruction from a tree-ring network, even if this climate variable is not the most important limiting factor on radial growth. In our case, May to August precipitation was the most important factor, and the second one was March-April TMP for almost all the chronologies. Classical approach in Dendroclimatology, is to use the PC 1 and/or high order PCs reconstruct precipitation. But here, we would like show that PC 1 could be a signal for precipitation but a noise for temperature. On the other hand the other PC's, which explain less variance, could be noise for precipitation and but a signal for temperature.

Specific Comments:
1. Page 3, line 52-53. Hughes et al., (2011) did not develop any reconstruction, but they investigated the climate signal.
Thank you for your attention we will correct it in the manuscript.

2. Page 5-lines 90-91: The authors should cite the investigators produced the chronologies.
We cited the investigators produced the chronologies.

3. Page 6-lines 128-129: What the authors mean by "Third, the final reconstruction is based 128 on bootstrap regression (Till and Guiot, 1990), the best method to assess the quality of the: : :" It is an awkward and not a scientific statement.
We will replace the sentence by: "Third, the final reconstruction is based on bootstrap regression (Till and Guiot, 1990), a method designed to calculate appropriate confidence intervals for reconstructed values and explained variance even in cases of short time-series."

4. Page 7-line 143: another awkward sentence "but bootstrap is much more interesting: : :"
We will replace by "… but bootstrap has the advantage to produce confidence intervals for such statistics without theoretical probability distribution and finally we accept the RE and CE for which the lower confidence margin at 95% are positive. This is more constraining than just accepting all positive RE and CE."

5. Pages 9-10, Temperature reconstruction: The authors mentioned that they conducted

PCA on the 23 chronologies. I have several questions and comments on this section. The authors used stepwise regression (SR), however, they did not give enough details about this procedure. I am concerned that the model could be over-fitted and some of the predictors could be just noise. What criteria were applied to end the stepwise process (e.g., p-to-enter, p-to remove)? Was a conventional statistic such as Mallows' Cp used to arrive at the final model? Does the validation CE and RE continue to increase through each step of the stepwise? Did the authors run SR on each calibration period independently or use the same variables that were suggested by the SR for the whole period?

We added information in the text under the titles "Data and Method", "Temperature reconstruction" explaining which method we used stepwise regression. We combined forward selection with backward elimination, checking for entry, then removal, until no more variables can be added or removed. Each procedure requires only that we set significance levels (or critical values) for entry and/or removal. We used p≤0.05 as entrance tolerance and p≤0.1 as exit tolerance. Actually, for almost all PCs it was p≤0.01 in entire regression. The final model obtained when the regression reaches a local minimum of RMSE. We also calculated Mallows Cp values. See the relation Cp and p (the number of parameters in the model, including the intercept) in (Figure1) .

[Figure]

We did not used a split-sample procedure to verify the model stability. We used bootstrap method. Therefore we run SR for the whole period. Bootstrap is only applied to the selected set of predictors by stepwise regression. Then it is not concerned by the bootstrap. We did not calculated calulated RE, CE at each step of the stepwise regression.

6. Page 7, section March-April temperature reconstruction. How did the chronologies cluster around each pc that they used in their equation?

We added a column to Table 3, to show the chronologies represented by higher magnitudes of the eigenvectors.

| | Explained variance (%) | Correlation coefficients with | | The chronologies represented by higher magnitudes** in the eigenvectors |
|---|---|---|---|---|
| | | May–August PPT | March–April TMP | |
| PC1 | 46.57 | 0.65 | 0.19 | KAR, KIZ, TEF, BON,USA,TUR, CAT, INC, ERC, YAU, SAV, TAN, SIU |
| PC2 | 7.86 | –0.07 | 0.15 | KAR, SAV, TIR, BOL, YAU, ESK, TEF,BON, SIU |
| PC3* | 4.93 | 0.04 | –0.48 | HCR, PAY, BOL, YAU, SIA |
| PC4* | 4.68 | 0.11 | 0.17 | TEF, KEL, FIR, SIA, KIZ, SIU, ART |
| PC5* | 4.42 | –0.25 | 0.27 | SAH, TIR, FIR, ART |
| PC6 | 3.73 | 0.15 | –0.14 | KIZ, FIR, SAV, KAR, TIR, PAY, ESK, TEF, BON, ART |
| PC7* | 3.56 | 0.19 | 0.18 | KIZ, BON, BOL, YAU, HCR, PAY, INC |
| PC8 | 2.87 | 0.26 | 0.01 | HCR, ESK, BON, FIR, ERC, SIA |
| PC9* | 2.45 | 0.16 | 0.17 | PAY, USA, BOL, YAU, TIR, HCR, FIR, SIA, SIU |
| PC10* | 2.21 | 0.14 | –0.08 | TUR, CAT, SAV, SIA, KEL, ERC, SIU |
| PC11 | 2.09 | –0.36 | –0.20 | HCR, TEF, USA, INC, PAY, TUR, SAV, SIU |
| PC12 | 1.80 | –0.12 | 0.05 | TEF, CAT, YAU HCR, ESK, USA, BOL, SIA |
| PC13 | 1.63 | –0.06 | 0.17 | TEF, TUR, BOL, KAR, YAU, SIA |
| PC14 | 1.55 | –0.14 | 0.06 | TIR, USA, FIR, TUR, YAU, KAR, BON |
| PC15* | 1.50 | –0.20 | –0.14 | KIZ, BON, USA, ESK, INC, BOL |
| PC16 | 1.31 | 0.04 | 0.08 | SAH, HCR, INC, YAU, SAV, KAR, FIR, BOL, SIU |
| PC17* | 1.25 | 0.15 | 0.19 | SAH, SIU, KAR, ESK, TUR, ERC |
| PC18 | 1.14 | 0.13 | 0.02 | KAR, TEF, TUR, SAV, BON, CAT |
| PC19 | 1.09 | 0.16 | –0.11 | PAY, INC, SAV, HCR, KEL, CAT, TAN |
| PC20 | 0.95 | –0.15 | –0.01 | TIR, SAH, CAT |
| PC21* | 0.89 | 0.06 | –0.28 | TUR, INC, TIR, SAV |
| PC22 | 0.85 | 0.44 | 0.10 | KIZ, SAH, BON, YAU, SIU |
| PC23 | 0.67 | –0.22 | –0.02 | TAN, KEL, TUR, CAT |

"*" indicates the PCs, which used in the reconstruction as predictors

"**" which exceed ±0.2 value.

7. Conclusion, line 325-327: it is an awkward statement. Did any of previous authors indicated in any of their publications it is IMPOSSIBLE to reconstruct the temperature in the eastern Mediterranean. Did the authors read the mind of these authors?

We tried to say with this sentence that no temperature reconstruction has been made, which mean that it is difficult to do that.

8. Show the minor ticks in Figure 4, 5, and 7.
We did suggested changes in the figures.

---

## Author Response (AR1)

Dear Dr. Luterbacher,

My coauthors and I thank you for your invitation to revise our manuscript. Here we will comment on, one-by one, the referee comments/suggestions.

Sincerely,
Nesibe Köse

**Response to RC1 comments**

Thank you for your time and comments. We would like to thank you for your time and comments.

General comments:

We could not use only the chronologies that have significant relationship to temperature, because at the same time they have significant precipitation signal (except ART chronology, Figure 2). On the other hand, we would like to show that it is possible to make a climate reconstruction from a tree-ring network, even if this climate variable is not the most important limiting factor on radial growth. In our case, May to August precipitation was the most important factor, and the second one was March-April TMP for almost all the chronologies. Classical approach in Dendroclimatology, is to use the PC 1 and/or high order PCs reconstruct precipitation. But here, we would like show that PC 1 could be a signal for precipitation but a noise for temperature. On the other hand the other PC's, which explain less variance, could be noise for precipitation and but a signal for temperature.

Specific Comments:
1. Thank you for your attention we corrected it in the manuscript.

2. We cited the investigators produced the chronologies.

3. We replaced the sentence by: "Third, the final reconstruction is based on bootstrap regression (Till and Guiot, 1990), a method designed to calculate appropriate confidence intervals for reconstructed values and explained variance even in cases of short time-series."

4. We will replace by "… but bootstrap has the advantage to produce confidence intervals for such statistics without theoretical probability distribution and finally we accept the RE and CE for which the lower confidence margin at 95% are positive. This is more constraining than just accepting all positive RE and CE."

5. We added information in the text under the titles "Data and Method", "Temperature reconstruction" explaining which method we used stepwise regression. We combined forward selection with backward elimination, checking for entry, then removal, until no more variables can be added or removed. Each procedure requires only that we set significance levels (or critical values) for entry and/or removal. We used $p \leq 0.05$ as entrance tolerance and $p \leq 0.1$ as exit tolerance. Actually, for almost all PCs it was $p \leq 0.01$ in entire regression. The final model obtained when the regression reaches a local minimum of RMSE. We also calculated Mallows Cp values. See the relation Cp and p (the number of parameters in the model, including the intercept) in (Figure1) .

[Figure]

We did not used a split-sample procedure to verify the model stability. We used bootstrap method. Therefore we run SR for the whole period. Bootstrap is only applied to the selected set of predictors by stepwise regression. Then it is not concerned by the bootstrap. We did not calculated RE, CE at each step of the stepwise regression. But based on the suggestion of both reviewer, for additional verification we also give split-sample procedure results using the same variables that were suggested for the whole period.

6. We added a column to Table 3, to show the chronologies represented by higher magnitudes of the eigenvectors.

7. We tried to say with this sentence that no temperature reconstruction has been made, which mean that it is difficult to do that.

8. We did suggested changes in the figures.

**Response to RC2 comments:**

Thank you for your time and valuable comments. We would like to thank you for your time and comments.

1. We give detailed information: "…….to produce time series with a strong common signal and without biological persistence. The residual chronologies may be more suitable to understand the effect of climate on tree-growth, even if any persistence due to climate might be removed by pre-whitening. ……. In this research we used residual chronologies obtained from ARSTAN to reconstruct temperature.

2. We added suggested information "Principle Component Analysis (PCA) was done over the entire period in common to the tree-ring chronologies. The significant PCs were selected by stepwise regression. We combined forward selection with backward elimination setting p≤0.05 as entrance tolerance and p≤0.1 as exit tolerance. The final model obtained when the regression reaches a local minimum of RMSE. The order of entry of the PCs into the model was $PC_3$, $PC_{21}$, $PC_4$, $PC_{15}$, $PC_5$, $PC_{17}$, $PC_7$, $PC_9$, $PC_{10.}$"

3. We replaced the sentence by: "Third, the final reconstruction is based on bootstrap regression (Till and Guiot, 1990), a method designed to calculate appropriate confidence intervals for reconstructed values and explained variance even in cases of short time-series." We calculate RE and CE values for 1901-29 and obtained low values. Therefore we removed discussion this part from the text and figure. For additional verification we present split calibration/verification results, which you mentioned that it may provide a more religious test, for the period 1930-2002 in Table 5.

4. Eq. (1) was removed as you suggested.

The suggested reference was added.

[revised manuscript text omitted]

---

## Referee Report (RR1)

**Referee report on "Spring temperature variability over Turkey since 1800 CE reconstructed from a broad network of tree-ring data"**

**Major comments:**

The article fills a gap among present tree-ring network studies and contains some novel features that merit its publication – but only after a revision and expansion of certain parts of the article. Normally, such short tree-ring temperature reconstructions as the present one (e.g. ~200 years) do not merit publication today, but since Turkey is a relatively under-sampled area – with a short instrumental record – an exception can be made here. What I, however, find perhaps even more interesting than the (short) temperature reconstruction presented in the article is the potential to present the chronologies as a "network study", looking at their different responses to various climate parameters over different seasonal windows. Such an analysis would make the article much more interesting and may provide valuable guidance for future research efforts. I would like to see the authors address/consider the following things before the article is published (although I understand that it might be too time consuming to address all of them fully):

1) A very interesting, and informative, addition to the work would be a network analysis of the Turkish tree-ring width records presented, preferably together with nearby ITRDB records, regarding their response to temperature and precipitation, respectively, during various seasonal windows. "Blue prints" to such a type of simple, yet very informative analysis, may be found in St. George (2014) and St. George and Ault (2014) and additionally in Hellmann et al. (2016). This information, preferably in the form of maps, could easily be included in a Supplement although a map of correlation to temperature and precipitation, respectively, during the key seasons considered could also be included in the main article.

2) The temperature signal in the tree-ring chronologies presented in the article is very weak and precipitation seems to be an even more important factor controlling the tree-ring growth. The PC analysis performed in the article requires the relationship between temperature and precipitation control to be relatively stable back in time (prior to the calibration period). This is very likely not the case. In many semi-arid areas the relative influence of temperature vs. precipitation control of tree-ring growth is highly unstable in time. Often the relationship between temperature and precipitation changes over time and there may occur dry and warm periods as well as dry and cold and likewise wet and warm as well as wet and cold periods. While I think this will be very difficult to test with regard to the Turkish tree-ring chronologies in the article the authors might at least be able to problematize this issue. I do not suggest that they should try to solve it but merely discuss it. One paragraph could easily be devoted to the issue and the problem could there be discussed in the light of previous tree-ring research in other semi-arid areas.

3) Given that the decorrelation decay length for temperature is in the order of a few thousand kilometres it might be a useful approach – since the Turkish instrumental records are so short – to look at the temperature variability further back in time from longer instrumental records in other parts of the Mediterranean. However, since the climate in the eastern Mediterranean is very different from that in the western Mediterranean any such analysis needs to be done with care. One simple approach could be to visually compare and maybe correlate the temperature reconstruction to longer instrumental records from the (eastern) Mediterranean region as a way to validate the temperature reconstruction prior to 1930.

4) It would be helpful if the authors would compare their new tree-ring based temperature reconstruction, and also the precipitation signal they find in the tree-ring chronologies, with existing gridded temperature and hydroclimate reconstructions for Europe. In Figure 2 the extracted local gridcell from the Old World Drought Atlas (Cook et al., 2015), the gridded seasonal precipitation reconstructions for Europe by Pauling et al. (2006), the gridded seasonal European temperature reconstructions by Luterbacher et al. (2004) and Xoplaki et al. (2005), and the new gridded European gridded summer temperature reconstruction by Luterbacher et al. (2016) could be used. A brief discussion could be added about what the local grid cells of these reconstructions show with regard to similarities/differences. It is important, however, to recognize that most of these products include limited data from Turkey.

5) There exists an extensive literature about (palaeo)climate conditions in the Mediterranean region but the authors hardly mention this research at all. In my opinion, it is a prerequisite for a scientific article to provide A) a general overview of the state-of-the-art knowledge in the field, and B) to place the new results obtained into a wider context of previous research in the same/similar field. I think a very useful start for the authors would be to consider the following works and the references cited there-in:

Lionello, P. (Ed.), 2012. The Climate of the Mediterranean Region, from the Past to the Future. Elsevier, Amsterdam, Netherlands.

Roberts, N. et al. 2012: Palaeolimnological evidence for an east-west climate see-saw in the Mediterranean since AD 900. Glob. Planet. Change, 84, 23–34.

Sanchez-López, G. et al. 2016: Climate reconstruction for the last two millennia in central Iberia: The role of East Atlantic (EA), North Atlantic Oscillation (NAO) and their interplay over the Iberian Peninsula. Quaternary Science Reviews 149: 135–150.

**Minor comments:**

Lines 45–47: This section about urban heat effect is rather interesting and could, preferably, be extended a little with references to other results, in other areas, of changes in the diurnal temperature range due to urbanization.

Lines 55–56: Also cite Cook et al. (2015) here.

Line 62: Change "the Medieval Climate Anomaly" to "the end of the Medieval Climate Anomaly".

Lines 123–126: Maybe add a short section of the sensitivity to the choice of gridded instrumental temperature product?

Line 289: It is an exaggeration to call it the "Great Depression".

**Literature cited/suggested:**

Cook, E. et al. 2015: Old World megadroughts and pluvials during the Common Era. Sci. Adv., 1, e1500561, doi:10.1126/sciadv.1500561.

Hellmann, L. et al. 2016: Diverse growth trends and climate responses across Eurasia's boreal forest. Environmental Research Letters: 11: 074021, doi:10.1088/1748-9326/11/7/074021.

Lionello, P. (Ed.), 2012. The Climate of the Mediterranean Region, from the Past to the Future. Elsevier, Amsterdam, Netherlands.

Luterbacher, J. et al. 2004: European seasonal and annual temperature variability, trends and extremes since 1500. Science, 303: 1499–1503.

Luterbacher, J. et al. 2016: European summer temperatures since Roman times. Environmental Research Letters, 11: 024001, doi:10.1088/1748-9326/11/1/024001.

Pauling, A. et al. 2006: Five hundred years of gridded high-resolution precipitation reconstructions over Europe and the connection to large-scale circulation. Clim. Dynam., 26: 387–405.

Roberts, N. et al. 2012: Palaeolimnological evidence for an east-west climate see-saw in the Mediterranean since AD 900. Glob. Planet. Change, 84, 23–34.

Sanchez-López, G. et al. 2016: Climate reconstruction for the last two millennia in central Iberia: The role of East Atlantic (EA), North Atlantic Oscillation (NAO) and their interplay over the Iberian Peninsula. Quaternary Science Reviews 149: 135–150.

St. George, S. 2014. An overview of tree-ring width records across the Northern Hemisphere, Quat. Sci. Rev. 95: 132–150.

St. George, S., and T. R. Ault. 2014. The imprint of climate within northern hemisphere trees, Quat. Sci. Rev. 89: 1–4.

Xoplaki, E. et al. 2005: European spring and autumn temperature variability and change of extremes over the last half millennium. Geophys. Res. Lett., 32, L15713, doi:10.1029/2005GL023424.

---

## Author Response (AR2)

Dear Dr. Luterbacher,

My coauthors and I thank you for your invitation to revise our manuscript. Here we will comment on, one-by one, the referee comments/suggestions.

Sincerely,
Nesibe Köse

**Response to RC1 comments**

Thank you for your time and comments, which improved our manuscript and gave us a chance to discuss our results in detail.

Major comments:

1. We added a correlation map as you suggested and discussed the spatial structure of climate-growth relationship (Fig. 4).

2. We discussed the issue over the instrumental period.

3. Indeed to compare our reconstruction to longer instrumental records from eastern Mediterranean region, we visually compare and correlate with the European temperature reconstruction by Xoplaki et al. 2005 to validate prior to 1930. This gridded reconstruction gave the higher correlation than the other gridded reconstructions (which you suggested in comment 4) over Turkey. We used the mean of corresponding grid points from European spring temperature reconstruction over the study area (36–42º N, 26–38º E) to show how the correlation changed over past 200 years. We found significant correlation (0.35, $p < 0.10$) for the period of 1901–1929 which climatic records are very few over the region while available data has sufficient quality for most part of Europe (Fig. 10).

4. We compared our tree-ring based temperature reconstruction with existing gridded temperature reconstructions for Europe (Xoplaki et al. 2005, Luterbacher et al. 2016) and the Old World Drought Atlas (OWDA) (Cook et al. 2015) (Figure 9a, b, c). We also compared the precipitation signal (PC1) obtained from our tree-ring network with Old World Drought Atlas (OWDA) (Cook et al. 2015) and gridded European summer precipitation reconstruction (Pauling et al., 2006) to test the strength of the signal spatially (Fig. 9d and e).

5. We added a review from suggested papers to Introduction.

Minor comments:

We cited Cook et al. 2015

We changed "the Medieval Climate Anomaly" to "the end of the Medieval Climate Anomaly".

We deleted "Great Depression".

We added "high resolution" to explain sensitivity of the gridded instrumental temperature.

[revised manuscript text omitted]

---

## Author Response (AR3)

Dear Dr. Luterbacher,

My coauthors and I thank you for your invitation to revise our manuscript. Here we will comment on, one-by one, the referee comments/suggestions.

Sincerely,
Nesibe Köse

1. For Figure 9: we calculated the field significance and plot the significant areas on the maps. For Figure 8, almost all areas were significant. So, we added a short text to Figure capture about significant areas.
2. We added more text to the capture of Figure 4 (now 3) and 10.
3. We created anew figure for Figure 6.
4. We included a text to conclusion part concluding comparison with existing spatial reconstructions.
5. We changed the Abstract based on your suggestions.

[revised manuscript text omitted]

---

## Author Response (AR4)

Dear Dr. Luterbacher,

My coauthors and I thank you for your invitation to revise our manuscript. Here we will comment on, one-by one, the referee comments/suggestions.

Sincerely,
Nesibe Köse

1. We recalculated significant fields using Monte Carlo Technique and changed the related figure and text based on the results.
2. We corrected the cited reference in the text and References. We improved the Introduction to make it more clear and logical.
3. Fig.3. We've redraw a clear figure.
4. Fig.4: We have already mentioned significance in fig. caption "Black lines surrounding circles represent significant correlation coefficients ($p < 0.05$).
5. Fig.5: We deleted the last sentence. We added a trend line for instrumental data on the figure.
6. Fig 6: We deleted the last sentence and added the information "does not include instrumental data"
7. You are right. Fig 7 and related text is confusing. Therefore we decided to remove Fig. 7 and the related text from the manuscript.
8. We've created a new figure for comparison of Xoplaki et al 2005 and TMP reconstruction by this study using standardized TMP inside of $C^0$. We corrected as Xoplaki et al 2005 on the figure. We also made necessary changes in the figure capture.
9. We have corrected the journal name.
10. We've corrected as "lower"